# Providing quality-assessed and standardised soil data to support global mapping and modelling (WoSIS snapshot 2023)

Niels H. Batjes[1], Luis Calisto[1], Luis Moreira de Sousa[1]

[1]ISRIC – World Soil Information, Droevendaalsesteeg 3, 6708 PB Wageningen, The Netherlands

*Correspondence*: Niels H. Batjes (niels.batjes@isric.org)

**Abstract.** Snapshots derived from the World Soil Information Service (WoSIS) are served freely to the international community. These static datasets provide quality-assessed and standardised soil profile data that can be used to support digital soil mapping and environmental applications at broad scale levels. Since the release of the preceding snapshot in 2019, refactored ETL (Extract, Load, Transform) procedures for screening, ingesting and standardising disparate source data have been developed. In
conjunction with this, the WoSIS data model was overhauled making it compatible with the ISO 28258 and Observations and Measurements (O&M) domain models. Additional procedures for querying, serving, and downloading the publicly available standardised data have been implemented using open software (e.g. GraphQL API). Following up on a short discussion of these methodological developments we discuss the structure and content of the "WoSIS 2023-snapshot". A range of new soil datasets was shared with us, registered in the ISRIC World Data Centre for Soils (WDC-Soils) data repository, and subsequently processed
in accordance with the licences specified by the data providers. An important effort has been the processing of forest soil data collated in the framework of the EU-HoliSoils project. We paid special attention to the standardisation of soil property definitions, description of the soil analytical procedures, and standardisation of the units of measurement. The "2023 snapshot" considers the following soil chemical properties (total carbon, organic carbon, inorganic carbon (total carbonate equivalent), total nitrogen, phosphorus (extractable-P, total-P, and P-retention), soil pH, cation exchange capacity, and electrical conductivity) and physical
properties (soil texture (sand, silt, and clay), bulk density, coarse fragments, and water retention), grouped according to analytical procedures that are operationally comparable. Method options are defined for each analytical procedure (e.g. pH measured in water, KCl or $CaCl_2$ solution, molarity of the solution, and soil/solution ratio). For each profile we also provide the original soil classification (i.e. FAO, WRB and USDA system with their version) and pedological horizon designations as far as these have been specified in the source databases. Three measures for "fitness-for-intended-use" are provided to facilitate informed data use:
a) positional uncertainty of the profile's site location, b) possible uncertainty associated with the operationally defined analytical procedures, and c) date of sampling. The most recent (i.e. *dynamic*) dataset, called *wosis_latest*, is freely accessible via various webservices. To permit consistent referencing and citation we also provide a *static* snapshot (in casu, December 2023). This snapshot comprises quality-assessed and standardised data for 228k geo-referenced profiles. The data come from 174 countries and represent more than 900k soil layers (or horizons) and over 6 million records. The number of measurements for each soil

property varies (greatly) between profiles and with depth, this generally depending on the objectives of the initial soil sampling programmes. In the coming years, we aim to gradually fill gaps in the geographic distribution of the profiles, as well as in the soil observations themselves, this subject to the sharing of a wider selection of "public" soil data by prospective data contributors; possible solutions for this are discussed. The WoSIS 2023-snapshot is archived and freely available at https://doi.org/10.17027/isric-wdcsoils-20231130 (Calisto et al., 2023).

## 1    Introduction

The World Soil Information service (WoSIS)  draws on a large complement of soil profile data that have been shared by numerous data providers. Nonetheless, a large proportion of the 800k "so-called" freely available soil profiles (see Arrouays et al., 2017), in practice, still remain "inaccessible" due to various licence constraints (e.g. Cornu et al., 2023). Soil data submitted for consideration in WoSIS come from a wide range of legacy holdings (e.g. traditional soil surveys), and increasingly include data derived from proximal sensing (e.g. Shepherd et al., 2022; Viscarra Rossel et al., 2016). The source data come in various formats and were determined according to a range of field sampling and soil analytical procedures, requiring standardisation and harmonisation during their ingestion/processing into WoSIS.

Prior to discussing the "2023 snapshot", we provide a short retrospective of activities that lead to the development of WoSIS. In the early days of desktop computers, ISRIC with its partners compiled a range of project-specific databases such as ISIS (van de Ven and Tempel, 1994), created to manage data for the ISRIC World Soil Reference Collection, several national and continental scale Soil and Terrain (SOTER) databases (e.g. FAO and ISRIC, 2003; FAO et al., 2007; FAO et al., 1998), the WISE database (Batjes, 1997; Batjes and Bridges, 1994), and the Africa Soil Profile (AfSP) database (Leenaars et al., 2014). While these different databases were structured along the general principles and criteria of the FAO Guidelines for Soil Description (FAO, 1977; 2006) and USDA Soil Survey Manual (Soil Survey Division Staff, 1993), the ISIS, SOTER, WISE, and AfSP databases each had their own data models and conventions. Further, out of necessity at the time, the databases were developed and implemented on stand-alone computers using a range of commercial software packages. In 2009, ISRIC management decided to bring the above stand-alone products together in a centralised enterprise database, known as WoSIS (World Soil Information Service), developed using PostgreSQL with the PostGIS extension for handling spatial data. After the initial ingest and standardisation of the above "ISRIC holdings" the service was to be expanded with datasets shared by a diverse range of soil data providers.

The original aim of WoSIS was to accommodate any type of soil data (profile, vector and grid) (Ribeiro et al., 2015; Tempel et al., 2013). However, from 2015 onwards, in view of technical considerations and institutional developments, the scope of WoSIS was changed to "safeguarding, processing, standardizing and serving geo-referenced soil profile (point) data for the

world" (Ribeiro et al., 2020). Alternatively, vector and grid maps derived from traditional soil mapping (e.g., Batjes, 2016; Dijkshoorn et al., 2005; FAO et al., 2012; van Engelen et al., 2006) and digital soil mapping (e.g., Hengl et al., 2017; Poggio et al., 2021; Turek et al., 2022) would be managed and served through other components of our spatial data infrastructure, such as the ISRIC data hub (https://data.isric.org) and the SoilGrids/WoSIS portal (https://soilgrids.org; last access: 24 April 2024). All these web services were developed using free-and-open-software (FOSS).

The ultimate goal of WoSIS, like for related global data compilation activities (Baritz et al., 2017; de Sousa et al., 2019), is full data harmonisation (Batjes et al., 2020; Ribeiro et al., 2015; Ribeiro et al., 2020). According, to the Global Soil Partnership (GSP, Baritz et al., 2014), harmonisation involves "providing mechanisms for the collation, analysis and exchange of consistent and comparable global soil data and information" and considers the following domains: a) soil description, classification and mapping, b) soil analyses, c) exchange of digital soil data, and d) interpretations. In view of the breadth and magnitude of the task, as well as the limited availability of comparative "multiple analytical procedures" data sets as required for full harmonisation (Batjes, 2023; Bispo et al., 2021; van Leeuwen et al., 2022) we have limited ourselves to the standardisation of soil property definitions, soil analytical procedures descriptions, plausibility checks for soil observation values and the standardisation of measurement units for commonly required soil properties (see Appendix A). Importantly, users should always keep in mind that the source datasets themselves (e.g., Armas et al., 2023; NPDB, 2023; USDA-NCSS, 2021) will provide more detailed information than WoSIS albeit not in a consistent, globally standardised format.

This paper discusses methodological changes to the WoSIS workflow and new data additions since the release of the previous snapshot (Batjes et al., 2020). First, we describe the new data model, the refactored data screening/ingestion process and indicate how the "shared" data are being served to the user community upon their standardisation. Thereafter, we describe the actual data screening, quality control and standardisation process. Subsequently, we describe the spatial distribution of soil profile sites and list the number of soil observations represented in the "WoSIS 2023-snapshot" (hereafter referred to as 2023-snapshot). In conjunction with this, we provide three measures for "fitness-for-intended-use" of the standardised data and discuss possible limitations of the snapshot. Finally, following up on a discussion concerning the scope for "full data harmonisation" in WoSIS, future developments and possible constraints arising are outlined.

The naming conventions and standard units of measurement are listed in Appendix A, while the structure of the snapshot files is described in Appendix B. In Appendix C we list the number of sites by country and continent (Table C1) as well as their distribution by world terrestrial ecosystems (Table C2) and biomes (Table C3).

Soils are important providers of ecosystem services (FAO and ITPS, 2015). WoSIS-served data have been used for a range of applications, such as predictive soil property mapping (Guevara et al., 2018; Moulatlet et al., 2017; Nenkam et al., 2022; Poggio et al., 2021; Turek et al., 2023), space and time modelling of soil organic carbon stock change (Heuvelink et al., 2021), and a diverse range of environmental assessments (e.g., Hassani et al., 2024; Huang et al., 2024; Luo et al., 2021; Lutz et al., 2019; Maire et al., 2015; Sanderman et al., 2017; Sothe et al., 2022). For example, based on the "2016 snapshot" and "2019

snapshot" respectively, Ivushkin et al. (2019) mapped global soil salinity change, while Wang et al. (2024) analysed responses of soil organic carbon under warming across global biomes. Ultimately, such information can help to inform the global conventions such as the UNCCD (United Nations Convention to Combat Desertification) and UNFCCC (United Nations Framework Convention on Climate Change), so that policymakers and business leaders can make informed decisions about the environment, biodiversity, and human well-being at an appropriate scale level.

## 2    WoSIS data model and workflow

### 2.1    Workflow

The data model and workflow for acquiring, ingesting, processing and serving data as described in Ribeiro *et al.* (2020) was overhauled. This proved necessary as this procedure was essentially designed as a series of dataset-specific python and SQL scripts, which was adequate as long as WoSIS was still relatively small. However, in view of the rapidly growing population of "shared" soil data and overall complexity of the data model itself it proved necessary to implement a new, state-of-the-art ISO domain model (de Sousa, 2023; de Sousa et al., 2023), with re-factored ETL (extract, transform and load) procedures, to ultimately better serve our diverse user community in our capacity as World Data Centre for Soils (WDC-Soils).

The main stages of the new workflow are visualised in Fig. 1: a) Data providers share their data with ISRIC WDC-Soils, b) the submitted data sets with associated metadata are screened for "completeness of information provided" (e.g. the licence defining access rights and description of terms and units) and, once considered adequate, subsequently stored "as is" in the WDC-Soils data repository (see "ISRIC Admin" in Fig. 1); and c) the source datasets are imported into the new WoSIS PostgreSQL relational database (see Sect. 2.2), using re-factored ETL procedures (see Sect. 2.3). Step c) includes: c1) basic data quality assessment and control, c2) standardising descriptions for the soil analytical procedures and units of measurement, and c3) automated checks against plausibility limits for each soil observation, see Sect. 3 for details. Subsequently, d) distribution of the quality-assessed and standardised data via various services such as dashboards and WFS (OpenGIS web feature service) as well as a metadata catalogue service.

*Insert Figure 1 near here*

### 2.2    Data model

As indicated earlier, a new data model for WoSIS was developed, aligned where possible with the ISO 28258 domain model (de Sousa et al., 2023) and the GloSIS web ontology (Palma et al., 2024), both stemming from O&M (Observations and Measurement,

Cox and David, 2011), all the while preserving legacy data. Main features of interest are the dataset (describes source of data), site (geo-spatial location where a soil investigation took place) and profile (sequence of pedo-genetic horizons along the depth of the profile). The key modification vis à vis the previous data model (Ribeiro et al., 2020) is the conditioning of analytical methods to the observation (see https://git.wur.nl/isric/databases/wosis-docs, last access: 26 April 2024). Changes made to the database schema and data over time are tracked using a migration tool (https://github.com/graphile/migrate, last access 24 April 2024). It maintains a record of the history, state, and dependencies of the database, including the conversion to the new data model.

Special attention was paid to the succinct description of the analytical procedures (see c2 above) using seven database tables, as summarised below:

- `thes_method_value`: Thesaurus of values that match the keys used to define an analytical method. For example, "natural clod" for the "sample type" key for method "bulk density".

- `thes_method_key`: Thesaurus of keys used to define an analytical method. For example, "reported pH", "exchange solution" and "index cation".

- `thes_method_option`: Encodes the possible combinations of key-value pairs for each numerical observation on layers. Note that only a small sub-set of observations can be associated with particular method options.

- `method_source`: Analytical methods descriptions as defined in the respective source databases (i.e. prior to standardisation). This table was imported "as is" from the old data model (Ribeiro et al., 2020) with the addition of a synthetic primary key. The records in this table remain essential to identify the method referred by each result.

- `method_standard`: Distinguishes each source method by the particular observation to which it applies. It can be regarded as a standardised description of the source method. Each record corresponds to a collection of key-value pairs in the `method_option` table for a single observation. Results for numerical observations reference this table to identify the corresponding analytical method.

- `method_option_standard`: Defines a many-to-many relationship between the `method_standard` and `method_option` tables. It determines the exact collection of key-value option pairs that constitute a standard method. The standard method is a specialisation of the source method for a specific observation.

## 2.3 ETL procedure

Extract, Transform and Load (ETL) is a standardised, semi-automatic process that guides the data processor during the ingestion of new datasets. Re-factored ETL procedures were developed to align with the structure of the ISO data model. During the initial phase, newly shared datasets are submitted to a quick consistency check (i.e. format, data model, metadata and licence after which they are uploaded "as is" to a staging area in the WoSIS system. Subsequently, during the transform stage the uploaded datasets are parsed by the system. During this process validation and standardisation occurs (see Sect. 3.3 for details). In case of (possible) unconformities the system will generate descriptive messages that guide the data processor towards possible actions that may be needed to resolve the flagged unconformities. The data processor then needs to correct these issues in conformity with the requirements of the WoSIS procedure manual in steps guided by the system; in some cases the original data providers may need to be consulted. At the end of this phase, the cleaned and standardised data remain in the staging area for final verification by a soil expert. After this verification, the final stage of the ETL process, "Load", can start. This is a fully automated process during which the cleaned and standardised data are copied into the WoSIS database and subsequently removed from the staging area (note: the source data themselves are permanently preserved in the ISRIC data repository). The newly ingested data can now be used to create a range of WoSIS-derived products (e.g., *wosis_latest*, *wosis_internal*, and dashboards, see Fig.1) in accord with the licences and possible restrictions specified by the data providers.

## 2.4 Operational definitions

Soil characteristics, such as texture, bulk density and organic carbon content, are collated according to a wide range of procedures in different countries. For such incongruent data to be interpreted correctly during the ETL process, the procedures for their collection, analysis, and reporting need to be well documented and understood. Results can differ when different analytical procedures are used even though these procedures may carry the same name (e.g. clay, silt and sand size fraction) or concept (see Soil Survey Staff, 2011). This makes the inter-comparison of different datasets difficult if it is not known how these data were collected/analysed. Therefore we use "operational definitions", as defined by USDA Soil Survey Staff (2011), for soil properties that are linked to specific analytical procedures. To properly characterise the "pH of a soil", for example, we need information on sample pre-treatment, soil/solution ratio, and description of solution (e.g. $H_2O$, 1 M KCl, 0.02 M $CaCl_2$, or 1 M NaF). Soil pH measured in Sodium Fluoride (pH NaF), for example, provides a measure for the phosphorus (P) retention of a soil whereas pH measured in water (pH $H_2O$) is an indicator for soil nutrient status. Consequently, in WoSIS soil properties are named according to and defined by the analytical procedures and corresponding "method options", based on common practice in soil science (e.g. BDFIOD for "bulk density (BD), fine earth fraction (FI), oven dry (OD)"). The current list of soil properties standardised in WoSIS is described in Sect. 3.3.

## 2.5    Data provisioning

Upon completion of the semi-automated ETL process, the quality-assessed and standardised data are distributed freely through various channels (see Fig. 1) in accordance with the license agreements (see Sect. 2.6):

- As *wosis_latest* (dynamic) via WFS; the respective endpoints are catalogued at the ISRIC Data Hub. 5 (https://data.isric.org/geonetwork/srv/eng/catalog.search#/search?any=wosis_latest, last access: 26 April 2024)

- As "fixed" snapshots (in TSV format) with a unique digital object identifier (DOI) to permit consistent citation (https://data.isric.org/geonetwork/srv/eng/catalog.search#/search?any=wosis_snapshot, last access:26 April 2024).

- The contents of *wosis_latest* can also be visualised using a dashboard with some querying and zooming facilities (https://dashboards.isric.org/superset/dashboard/wosis_latest, last access: 26 April 2024).

- 10 Profile data from *wosis_latest* can also be queried through the "SoilGrids web platform" (https://soilgrids.org/, last access: 26 April 2024), which also provides access to a range of soil property maps derived from the WoSIS-served profile data and a set of environmental covariates using digital soil mapping (Poggio et al., 2021; Turek et al., 2023).

- The *wosis_latest* holdings can also be queried using a GraphQL interface (https://graphql.isric.org/, last access: 26 April 2024) that facilitates exploration of the data (e.g. select data for organic carbon, bulk density, proportion of coarse fragments 15 per layer (horizon) for profiles located in a given geography). Results of such tailor-made queries can then be exported as input in scripting languages such as Python or R (R Core Team, 2021), for example to calculate regional carbon stocks.

## 2.6    Licence agreements

It is not a simple task to find potential providers of "open" soil data (Arrouays et al., 2017; Batjes, 2009; Cornu et al., 2023). This 20 may be due to technical issues, access arrangements, reasons for sharing (e.g., "Why share the data and for what purpose? What is in it for us?"), as well as legal requirements (Bispo et al., 2021; Robinson et al., 2019). All data sets that are shared with our centre are first registered in the ISRIC Data Repository together with their metadata; data sharing agreements should align with the ISRIC Data Policy (ISRIC, 2016). During the subsequent WoSIS standardisation workflow, we are faced with three different types of datasets. First, those with a non-restrictive Creative Commons (CC-BY) licence, defined here as at least a CC-BY 25 (Attribution) or CC-BY-NC (Attribution Non-Commercial) licence (these are later served as *wosis_latest)*. Second, datasets with a more "restrictive" licence in the sense that they can exclusively be used for "visualisations", such as SoilGrids™ (i.e. *wosis_internal,* see Fig. 1), by ISRIC itself. The latter, generally because the coordinates cannot be disclosed as stipulated by certain data providers (for details see https://www.isric.org/explore/wosis/wosis-contributing-institutions-and-experts, last access: 26 April 2024). Finally, several data sets have licences that stipulate that they should only be safeguarded in the ISRIC 30 repository and cannot be used for any data processing (i.e. permanent embargo).

The number of profiles in WoSIS per licence category, i.e. "public" respectively "restricted", can be viewed and filtered using a dashboard (https://dashboards.isric.org/superset/dashboard/wosis_licenses/, last access: 26 April 2024). As shown in Table 1, the number of "public access" profiles served from WoSIS as snapshots increased from 96k in 2016 to 228k in 2023. Conversely, it should be noted here that a large proportion of the forest soil data "shared" in the framework of the EU-Holisoils project, for instance, could not be included in the "2023 snapshot" due to licence restrictions specified by the data providers. As a result, only 34k out of the total of 107k profiles "shared" with ISRIC between 2019 and 2023 could actually be included in the 2023-snapshot (resp. *wosis_latest*).

*Insert Table 1 near here*

## 3 Data screening, quality control and standardisation

### 3.1 Consistency checks

Soil profile data shared for possible consideration in WoSIS were sampled and analysed according to various national or international standards and presented in various formats (from paper to digital). They are of varying degree of completeness as discussed below. To be considered in the WoSIS standardisation workflow (Fig. 1), each soil profile must meet several criteria as described earlier in Batjes et al (2020, p. 301). Summarising, they must be associated to a site correctly geo-referenced, have consistently defined upper and lower depths for each layer (or pedogenetic horizon), and have observations for at least some of the soil properties that are being served (e.g. sand, silt, clay and pH) as well as a succinct description of the analytical procedures and units of measurement. A soil (taxonomic) classification is considered desirable though not mandatory. Profiles associated to a valid site, for which only the classification is specified in the source data can still be useful for mapping of soil taxonomic classes.

Consistency in layer depth (i.e. sequential increase of the upper and lower depth reported for each layer down the profile) is checked using automated procedures (see Sect. 3.2). In line with current internationally accepted conventions, such depth increments are given as "measured from the soil surface, including organic layers and mineral covers" (FAO, 2006; IUSS Working Group WRB, 2022; Schoeneberger et al., 2012; Soil Survey Staff, 2022b). Until 1993, however, the begin (zero datum) of the profile was set at the top of the mineral surface (the *solum* proper), except for "thick" organic layers as defined for peat soils (FAO, 1977; 1990). Organic horizons were recorded as above and mineral horizons recorded as below, relative to the mineral surface (Schoeneberger et al., 2012, p. 2-6). As far as possible, such "organic_surface" layers are flagged in the snapshot (see Appendix B) so that they may be filtered-out during auxiliary computations of soil organic carbon stocks, for example.

**3.2    Screening for duplicate profiles**

In the early stage of WoSIS, many source databases were compilations of shared soil profile data necessitating intricate procedures for identifying and flagging possibly repeated profiles (see Batjes et al., 2017; Ribeiro et al., 2020). Soil profiles located within 100 m of each other are flagged as possible duplicates, provided the year of sampling is identical (this criterion allows for reporting results of soil monitoring campaigns at the same *site*). Upon additional automated checks concerning the thickness of the first three soil layers (i.e. upper and lower depth), sand, silt and clay content, the duplicate profiles with the least-comprehensive component of observations are flagged and excluded from further processing (i.e. distribution). When still in doubt after these rigorous tests a final visual "similarity check" is made with respect to other commonly reported soil properties such as $pH_{water}$ and organic carbon content, possibly leading to the flagging (exclusion) of some additional profiles.

**3.3    Standardisation of property names, analytical procedure descriptions and units of measurement**

A crucial step during data ingestion is the standardisation of the, regularly non-English, soil property names used in the source databases to the WoSIS conventions, as well as the standardisation of the soil analytical procedures according to consistent "operational definitions" (see Appendix A). Subsequently, the units of measurement are standardised, and the reported measurement values assessed according to soil observation-specific plausibility ranges for the respective soil properties (i.e. likely minimum and maximum). Some of these plausibility limits may change when more data become available for so far under-represented soil observations, similar to ICP Forests (2020, p. 25), and appropriate PostgreSQL "trigger mechanisms" have been implemented for this. Data that do not meet these conditions are flagged and not processed further in the ETL workflow (see above), unless the observed "inconsistencies" can easily be solved (e.g. blatant typos in pH values). Alternatively, the data provider(s) may be contacted to resolve the observed errors.

Similar to the 2019-snapshot, the following soil properties are considered in the 2023-snapshot:

- Chemical: total carbon (i.e. organic plus inorganic carbon), organic carbon, inorganic carbon (i.e. total carbonate equivalent), total nitrogen, soil pH, cation exchange capacity, electrical conductivity, and phosphorus (extractable-P, total-P, and P-retention),
- Physical: Soil texture (clay, silt, sand), coarse fragments, bulk density, and water retention.

All measurement values are served as recorded in the source data, after the above consistency checks and standardisation of the units of measurement to the target units (see Appendix A). As such, we *do not* apply any "gap filling" procedures during ETL nor do we apply any pedotransfer functions (PTF) to derive missing bulk density data or soil hydrological properties, respectively

harmonise particle class size limits to a common standard, for example. This follow up stage of data processing is seen as the task of the data users (modellers) themselves. In practice, the required PTFs or ways for depth-aggregating the layer data will be determined by the projected use(s) of the standardised data (see Finke, 2006; Heuvelink et al., 2021; Poggio et al., 2021; Turek et al., 2023; van Leeuwen et al., 2024; Van Looy et al., 2017). It should be noted, however, that inadvertently some PTF-derived values (e.g. for bulk density) could have slipped through the above consistency checks in situations where procedures were mis-coded in the metadata of a source data set; critical modellers should exclude such values during their analyses.

### 3.4    Providing measures for fitness-for-intended-use

As indicated earlier, data served from WoSIS are used for a wide range of environmental applications(e.g., Guevara et al., 2018; Heuvelink et al., 2021; Luo et al., 2021; Maire et al., 2015 ; Moulatlet et al., 2017; Poggio et al., 2021; Sanderman et al., 2017; Sothe et al., 2022; Turek et al., 2023), but many of these assessments do not explicitly consider the uncertainties that are associated with the data. However, it is well known that "soil observations used for calibration and interpolation are themselves not error-free" (e.g., Baroni et al., 2017; Cressie and Kornak, 2003; Folberth et al., 2016; Grimm and Behrens, 2010; Guevara et al., 2018; Heuvelink, 2014; van Leeuwen et al., 2022). Therefore, since 2019, we provide three measures for "fitness-for-intended-use" in *wosis_latest* namely: a) positional uncertainty of the profiles (i.e. site location), b) inferred accuracy of the laboratory measurements, and c) date of sampling. These three measures, although approximative, should be duly considered in digital soil mapping and subsequent earth system modelling as they can affect the prediction uncertainty and "area-of-applicability" of the resulting derived products (Dai et al., 2019; Meyer and Pebesma, 2021; Shi et al., 2023). For example, large areas of the globe are still poorly represented in WoSIS (basically the yellow areas in Fig. 3). As indicated earlier, this issue can only be remedied when a larger selection of datasets is shared by the international soil community for consideration in WoSIS.

Importantly, prospective data users should also realise that the point/profile data shared for consideration in WoSIS are largely based on purposive sampling. During such "traditional" surveys, soil surveyors identify sample locations based on their knowledge of the survey area, desired level of detail (scale) and objective of the survey, for example detailed or exploratory surveys (FAO, 2006; IUSS Working Group WRB, 2022; Soil Survey Staff, 2017). Hence, such "legacy" data are not based on a probabilistic sampling scheme as recommended for digital soil mapping  (Brus et al., 2011; Brus, 2022; Cramer et al., 2019; Heuvelink et al., 2007).

### 3.4.1    Positional uncertainty

Profiles in WoSIS are georeferenced through the *site* in which they were sampled in accord with ISO 28258 standards (de Sousa et al., 2023). The coordinates themselves are presented according to the World Geodetic System datum ensemble (i.e. WGS84,

EPSG code 4326) upon their conversion from a diverse range of national projections. For most profiles (86 %, see Table 2) the approximate positional uncertainty of the profile locations, as inferred from the coordinates given in the source datasets, is ~100 m. Typically, geo-referencing before the advent of GPS (Global Positioning Systems) in the 1970s is less accurate; often we just do not know the "true" accuracy. Nonetheless, digital soil mappers should be aware of this issue (Grimm and Behrens, 2010),

because the soil observations and environmental covariates may not actually overlap (Cressie and Kornak, 2003), this both in space and time.

*Insert Table 2 near here*

### 3.4.2 Measurement uncertainty

Soil data managed in WoSIS have been analysed according to a diverse range of analytical procedures in multiple laboratories. A measure for measurement uncertainty is thus desired. Soil laboratory-specific Quality Management Systems and laboratory proficiency-testing (PT) can provide this type of information (GLOSOLAN, 2023; Magnusson and Örnemark, 2014; Munzert et al., 2007; NATP, 2015; WEPAL, 2019). Calculation of laboratory-specific measurement uncertainty for a single procedure, respectively multiple analytical procedures, will require several measurement rounds (years of observation) and solid statistical

analyses (van Leeuwen et al., 2022). Generally, however, this type of information is not provided with the source data sets submitted to the ISRIC data repository. Therefore, pragmatically, we have distilled the required information from the PT-literature (Al-Shammary et al., 2018; ICP Forests, 2021a; Kalra and Maynard, 1991; Rayment and Lyons, 2011; Rossel and McBratney, 1998; van Reeuwijk, 1983; WEPAL, 2019), as far as technically feasible. In the case of organic carbon content, for example, the mean variability was 17 % (with a range of 12 to 42 %) and for "CEC buffered at pH 7" of 18 % (range 13 to 25%) when multiple

laboratories analyse a standard set of reference materials using similar operational procedures (WEPAL, 2019).

The figures for measurement accuracy presented in Appendix A represent first approximations. They are derived from the inter-laboratory comparison of analyses on well-homogenised, reference samples for a still relatively small range of soil types. These indicatory figures should be refined, for example using probability distribution functions (Heuvelink et al., 2007; van Leeuwen et al., 2022), once sufficient laboratory and procedure-related accuracy (i.e. systematic and random error) information

is provided with the shared soil data (Magnusson and Örnemark, 2014). Alternatively, this type of information may be collated in the context of international laboratory PT-networks such as GLOSOLAN and WEPAL, and in the framework of the ongoing LUCAS topsoil monitoring round (Bispo et al., 2021; Cornu et al., 2023). Meanwhile, the present first estimates can already be considered when calculating the uncertainty of predictive digital soil maps and of any interpretations derived from them (e.g. studies of soil organic carbon stock change).

Realistically, full harmonisation of analytical data derived from disparate sources, the ultimate ambition in WoSIS, will first become feasible once results of a representative set of multi-procedure, inter-laboratory comparison data sets become (freely)

available, as discussed by Baritz *et al.* (2014), Bispo *et al.* (2021) and Batjes (2023), and a common set of reference Standard Operating Procedures (SOPs) has been accepted as a global standard.

### 3.4.3 Year of sampling

5 For each profile site, the date of sampling has been recorded as far as documented in the source data. This information is important to consider when superimposing the profile data with environmental co-variates, such as land cover, for example in the context of space and time analyses (Giller et al., 2006; Heuvelink et al., 2021). Most (54%) profiles represented in the snapshot were described/sampled between 1980 and 2020 (Table 3), and less than 4% before 1960. Alternatively, the date of site description and sampling is not known for almost 27% of the profiles as the information was not provided in the source materials.

*Insert Table 3 near here*

## 4 Spatial distribution of soil profiles and number of observations

### 4.1 Spatial distribution

15 The 2023-snapshot includes standardised data for 228k profiles, sampled at 217k different sites (Fig. 2). The greatest number of profiles come from north America (35 %) followed by Oceania (19%) and Europe (17%), while there are still few profiles for Asia (3%) and Antarctica (Table 4). The profiles come from sites in 174 countries. The average density of observations varies greatly both between countries (Table C1) and within each country.

Changes in the spatial distribution and density of profiles (per 1000 $km^2$) in the successive WoSIS snapshots (Fig. 3) reflect
20 the degree to which our data acquisition efforts were successful, as further discussed in Sect. 6. Overall, the density of soil observations is still low for Central Asia, Southeast Asia, Central and Eastern Europe, Russia, and the northern circumpolar region in the 2023-snapshot.

*Insert Figure 2 near here*

The number of profiles by biome (Olson et al., 2001b) and broad climatic region (Sayre et al., 2014) respectively, as derived from GIS overlays, are listed in Table C2 and C3.

*Insert Table 4 near here*

*Insert Figure 3 near here*

## 4.2     Number and depth of observations

In total, the profiles considered in the 2023-snapshot are described by 0.9 million soil layers (or horizons). This corresponds with
over 6.1 million records that include both numeric (e.g. silt content, soil pH, and cation exchange capacity) as well as class (e.g. WRB soil classification and horizon designation) properties. There are more observations for the chemical properties than the physical properties (see Table A1). Further, the number of observations generally decreases with depth, this largely depending on the objectives of the original soil surveys. The interquartile range (Q1-Q3) for maximum depth of soil sampled in the field is 33-150 cm, with a median (Q2) of 100 cm (mean= 107 cm). It should be noted here that most specific purpose surveys only
consider the topsoil (e.g. soil fertility surveys), while others systematically sample soil layers up to depths exceeding 20 m (with a maximum of 32 m). When data from such "specific purpose surveys" (defined here as < 30 cm and >300 cm) are excluded, the figures for maximum depth sampled become: Q1= 90 cm, Q2= 122 cm, Q3= 155 cm with a mean of 126 cm.

Table 5 provides an overview of the maximum depth of soil sampled during the various surveys that underpin WoSIS, by continent. Unfortunately, we are not able to show the "depth to bedrock" as this information is seldom made explicit in the source
databases.

*Insert Table 5 near here*

## 5     Distributing the standardised data

The standardised data are distributed through ISRIC's Spatial Data Infrastructure (SDI). The SDI is based on open source technologies and open web-services (WFS, WMS, WCS, CSW) following Open Geospatial Consortium (OGC) standards and

aimed specifically at handling soil data. Our metadata are organised following standards of the International Organization for Standardization (ISO-19139, 2019) using GeoNetwork (see https://data.isric.org). The WoSIS database is hosted in a PostgreSQL database, with the spatial extension PostGIS. The PostgreSQL database itself is connected to MapServer to permit data download from GeoNetwork. These processes are aimed at facilitating global data interoperability and citation in compliance with FAIR

principles. The data should be "findable, accessible, interoperable, and reusable" (Wilkinson et al., 2016).

*Static* snapshots are given a unique DOI (digital object identifier) to permit consistent citation. The 2023-snapshot is distributed in tab-separated values format (see Appendix B for file structure) and as GeoPackage (https://doi.org/10.17027/isric-wdcsoils-20231130). An online Readme file, that includes links to two short tutorials, provides additional technical information (https://www.isric.org/sites/default/files/Readme_WoSIS_202312_v2.pdf). Alternatively, the evolving *dynamic* version of the

standardised data (i.e. *wosis_latest)* can be accessed/queried through the ISRIC Data Hub (https://data.isric.org) and the SoilGrids platform (https://soilgrids.org). Tutorials describing how to access *wosis_latest* from QGIS using WFS and with GraphQL (Calisto, 2023) can be found on the ISRIC website (see https://www.isric.org/explore/wosis/faq-wosis, last access: 26 April 2024).

By its nature, the *dynamic* version will grow when new profile data are shared and processed, additional soil properties are

considered in the WoSIS workflow, and/or when possible corrections are required. Potential errors can be reported via a "Google group" (https://groups.google.com/forum/#!forum/isric-world-soil-information, last access: 26 April 2024) so that these may be addressed in the *dynamic* version.

## 6      Discussion

We describe new procedures for handling and standardising disparate world soil profile data in WoSIS. The data model was fully harmonised to ISO 25828 and O&M requirements, with minor adjustments, and refactored ETL procedures were implemented. Alternatively, it should be stressed that the ultimate, desired full harmonisation of observations to an agreed reference analytical procedure Y, for example "pH $H_2O$, 1:2.5 soil/water solution" for say all "pH 1:x $H_2O$" measurements, will first become feasible once the target procedure (Y) for analysing each property has been defined, and subsequently accepted as "global standard" by

the international soil community. A next step would be to collate/develop "comparative" data sets for each soil property (i.e. sets with samples analysed according to a given reference procedure ($Y_i$) and the corresponding national procedures ($X_j$)) for pedotransfer function development. These relationships, however, will often be soil type and region specific (GlobalSoilMap, 2015) and difficult to develop (i.e. calibrate and validate) when datasets for the comparisons do not yet exist or are simply not freely shared/available (Batjes, 2023; Bispo et al., 2021; Cornu et al., 2023; van Leeuwen et al., 2024). Hence the importance of

regional laboratory inter-comparison programmes, such as those undertaken in the framework of for example ANSIS (2023),

GLOSOLAN (2023), ICP Forests (2021a) and LUCAS (Bispo et al., 2021), that aim to develop consistent, context-specific (e.g. by country or land use/soil type) pedotransfer functions towards an agreed set of SOPs. However, it should be noted that the standard SOPs specified by these various programmes need not be comparable. In his context, Suvannang *et al.* (2018) observed that "comparable and useful soil information (at the global level) will only be attainable once laboratories agree to follow common

standards and norms". Over the years, however, many organisations/countries have implemented analytical procedures and quality assurance systems that are well suited for their specific purposes (e.g., ANSIS, 2023; Cornu et al., 2023; Orgiazzi et al., 2018; Soil Survey Staff, 2022a). Consequently, they may not be inclined to harmonise their data to a (still to be decided) set of global "reference" SOPs. However, agreed upon procedures for such a full scale harmonisation will be required when developing a globally federated, and ultimately interoperable, spatial soil data infrastructure (GLOSIS, de Sousa et al., 2021) through which

(pre-harmonised) source data are served and updated by the respective data providers, and made queryable according to a common standard (de Sousa et al., 2023; OGC, 2019).

It is our intention to gradually fill gaps in the geographic distribution (Fig. 3) and range of soil properties (Appendix A) in the coming years. This work is part of ISRIC's remit as a regular member of the World Data System (https://worlddatasystem.org, last access: 26 April 2024). The degree to which this will be feasible, however, will largely depend on the willingness and ability

of data providers to share (some of) their data for consideration in WoSIS. For the northern Boreal and Arctic region, for example, ISRIC can draw on new profiles collated by the International Soil Carbon Network (ISCN, see Malhotra et al., 2019). Alternatively, it should be reiterated that several datasets in our repository (e.g. ICP Forests, 2021a) can *only* be standardised and used for SoilGrids$^{TM}$ applications due to existing licence restrictions. Conversely, some countries such as New Zealand distribute their national soil profile dataset with a CC-BY-ND 4.0 licence, which implicitly precludes making any derivatives and hence

their standardisation in WoSIS (see https://viewer-nsdr.landcareresearch.co.nz/datasets/downloads/1042-2, last access 10 June 2024).

Concerning the actual scope for expanding *wosis_latest* in the coming years, we noted that getting positive responses to our requests for sharing soil data is becoming increasingly cumbersome; the overall success rate during the "2019-2023" acquisition effort was around 25%. However, many of these datasets are being "shared" with ISRIC with the provision that the profile

coordinates themselves may not be shown; hence, the corresponding soil data cannot be "openly" served to our user community through *wosis_latest*. Further, the site and profile coordinates are then regularly shared as "theoretical coordinates" only (e.g. ICP Forests, 2021b; Poeplau et al., 2020), highlighting the need for considering positional uncertainty in digital soil mapping and other applications. Another source of concern is that major soil monitoring programmes, such as LUCAS (e.g. Ballabio et al., 2016; Orgiazzi et al., 2018), only consider the top 20 or 30 cm of the soil. That is, they do not consider the actual soil profile

depth as required for more comprehensive soil assessments such as computing changes in global carbon stocks or mapping plant-available water holding capacity in the root zone (e.g.  Batlle-Bayer et al., 2010; Leenaars et al., 2018; von Haden et al., 2020; Wang et al., 2022).

**7        Data availability**

The 2023-snapshot is archived for long-term storage at ISRIC – World Soil Information, the World Data Centre for Soils (WDC-Soils) of the ISC (International Council for Science) World Data System (WDS). It is freely accessible at https://doi.org/10.17027/isric-wdcsoils-20231130 (Calisto et al., 2023). The zip file (446 Gb) includes a copy of the "readme"

file, the data in TSV format (see Appendix B) respectively in OGC GeoPackage format.

**8        Conclusions**

Bringing disparate soil profile data from different sources under a common global standard poses many and diverse challenges. A major improvement has been the harmonisation of the WoSIS data model to ISO 28258 and O&M domain specifications. In

conjunction with this re-factored ETL procedures greatly improved the data ingestion and standardisation process, and new ways for visualising, querying and serving the data were developed to better serve our user community.

There are still numerous gaps in terms of geographic distribution as well as range of soil taxonomic units and/or soil properties represented. We aspire to address such gaps in future updates of '*wosis_latest*'. However, as World Data Centre, we are largely dependent on the ability of soil data owners to share some of their data freely for the greater benefit of the international

community. To facilitate and stimulate this process, we are developing a web-based facility (front-end) to permit data providers to directly upload their soil data to WoSIS in a consistent format based on the refactored ETL procedures. As an incentive, upon their standardisation, we aim to provide each data provider with a tailor-made dashboard for viewing and querying the datasets they shared, possibly with a DOI to facilitate citation.

Various sources of uncertainty are associated with the data. Therefore, we provide three measures for "fitness-for-intended-

use" of the standardised data. This information, although coarse, should be duly considered by prospective users of the snapshot.

Unfortunately, numerous soil datasets worldwide are not freely accessible for various reasons. Standardised procedures, mechanisms, policies and incentives aimed at encouraging soil data sharing by different categories of data owners/providers are needed (e.g., Fantappie et al., 2021; Gobezie and Biswas, 2023; Padarian and McBratney, 2020; Robinson et al., 2019). At a transnational level, these pressing and complex issues are being addressed by the Global Soil Partnership, hosted by UN-FAO,

in the context of the evolving federated Global Soil Information System.

**Appendix A: Coding convention**s

*Insert Table A1 here*

                           *Insert Table A2 near here*

**Appendix B: Structure of WoSIS 2023-snapshot**

This Appendix describes the structure of the data files served with the WoSIS 2023-snapshot, namely

10   *wosis_202312_observations.tsv*,   *wosis_20312_site.tsv,*   *wosis_202312_profiles.tsv*,   *wosis_202312_layers.tsv*,   and *wosis_202312_xxxx.tsv* (where "*xxxx*" is the name of the observation). The data files are also distributed as OGC GeoPackage, which stores the files within an SQLite database. Technical details are provided in a Readme file (https://www.isric.org/sites/default/files/Readme_WoSIS_202312_v2.pdf).

15   • *wosis_202312_observations.tsv*: This file lists the four to six letter codes for each observation, whether the observation is for a site/profile or layer (horizon), the unit of measurement and the number of profiles respectively layers represented in the snapshot. It also provides the inferred accuracy for the laboratory measurements (see Appendix A).

| | |
|---|---|
| code | Code for the observation |
| property | Description of soil property |
| 20    procedure | Description of analytical procedure |
| unit | Standard unit of measurement |
| profiles | Number of profiles that have at least one measurement for the observation |
| layers | Number of layers that have measurements for the observation |
| accuracy | Inferred accuracy of the laboratory measurements (First approximation, see Sect. 3.4.2) |

• *Wosis_202312_site*.tsv: This file characterises the site location where profiles were sampled. The following field names are used:

site_id                         Primary key

| | longitude | Longitude in degrees (WGS84) |
|---|---|---|
| | latitude | Latitude in degrees (WGS84) |
| | positional_uncertainty | Positional uncertainty of the profile's site location, expressed in four classes (see Table 2) |
| 5 | country_name | Name of country where site is located |
| | region | Region in which site is located |
| | continent | Continent in which site is located |

10 • *wosis_202312_profiles.tsv*: Presents the unique profile ID (i.e. primary key), site_id, source of the data, country ISO code and name, positional uncertainty, latitude and longitude (WGS 1984), maximum depth of soil described and sampled, as well as information on the soil classification system and edition. Depending on the soil classification system used, the number of fields will vary. For example, for the World Soil Reference Base (WRB) system the options are publication year (i.e. version), reference_soil_group_code, reference_soil_group_name, and the name(s)

15 of the prefix (primary) qualifier(s) respectively suffix (supplementary) qualifier(s). The terms principal qualifier and supplementary qualifier are used since 2015 (IUSS Working Group WRB, 2015; 2022); earlier WRB versions used prefix and suffix for this (e.g. IUSS Working Group WRB, 2006). Alternatively, for USDA Soil Taxonomy, the version (year), order, suborder, great group, and subgroup can be accommodated (Soil Survey Staff, 2014). The following field names are used:

| | | |
|---|---|---|
| 20 | profile_id | Primary key |
| | profile_code | Code for the profile |
| | dataset_code | Identifier for source data set |
| | site_id | Identifier for site where profile is located |
| 25 | positional_uncertainty | Positional uncertainty of the profile's site location, expressed in four classes (see Table 2). |
| | country_name | Name of country where site is located. |

| | | |
|---|---|---|
| | longitude | Longitude in degrees (WGS84) |
| | latitude | Latitude in degrees (WGS84) |
| | wrb_reference_soil_group_code | Code for WRB group (in given version of WRB) |
| | wrb_reference_soil_group | Full name for reference soil group |
| 5 | wrb_prefix_qualifiers | Name for prefix (i.e. for WRB1988) |
| | wrb_suffix_qualifiers | Name for suffix (i.e. for WRB1988) |
| | wrb_principal_qualifiers | Name for principal qualifiers (i.e. for WRB 2015 and WRB 2022) |
| | wrb_supplementary qualifiers | Name for supplementary qualifiers (i.e. for WRB 2015 and WRB 2022) |
| | wrb_publication_year | Version of World Reference Base for Soil Resources |
| 10 | fao_major_group_code | Code for major group (in given version of the Legend), |
| | fao_major_group | Name of major group |
| | fao_soil_unit_code | Code for soil unit |
| | fao_soil_unit | Name of soil unit |
| | fao_publication_year | Version of FAO Legend (e.g. 1974 or 1988) |
| 15 | usda_order_name | Name of USDA Soil Taxonomy order |
| | usda_suborder | Name of USDA Soil Taxonomy suborder |
| | usda_great_group | Name of USDA Soil Taxonomy greatgroup |
| | usda_subgroup | Name of USDA Soil Taxonomy subgroup |
| | usda_publication_year | Version of USDA Soil Taxonomy |

- W*osis_202312_layers.tsv*. This file characterises the layers (or horizons) per profile:

      profile_id                          Primary key

      layer_id                            Sequential number for the layer (or horizon)

      profile_code                     Code for the profile

site_id                              Identifier for site where profile is located

      layer_name                    Name of pedogenetic horizon ("as is")

      upper_depth                    Upper depth of layer

      lower_depth                    Lower depth of layer

      layer_number                  Sequential number for the layer (or horizon)

organic_surface              Flag for the presence of an organic layer above the mineral soil

      dataset_id                      Abbreviation for source data set (e.g. WD-ISCN)

      licence                          Licence for observation as indicated by the data provider (e.g. CC BY)

- *Wosis_202312_xxxx.tsv*. For each observation (e.g. "xxxx" = "BDFIOD"), as defined under "code" in file *wosis_202312_observation.tsv,* the following are listed:

profile_id                         Primary key

      layer_id                            Primary key (number, sequential from top to bottom)

      profile_code                     Code for given profile

      layer_name                    Name of pedogenetic horizon ("as is")

      upper_depth                    Upper depth of layer

lower_depth                    Lower depth of layer

| | |
|---|---|
| organic_surface | Indicates if there is an organic layer above the mineral surface |
| value | Array listing all measurement values for observation "*xxxx*" for the given layer. In some cases, more than one observation is reported for a given horizon (layer) in the source, for example four values for TOTC: [1:5.4, 2:8.2, 3:6.3, 4:7.7 ] (see value_avg below) |
| method_options | Array listing the method options for each analytical procedure as distilled from the source data. The content of this array varies with the soil observation under consideration as described in the method option table for each analytical procedure. For example, in the case of electrical conductivity (ELCO), the method options include sample pretreatment (e.g. sieved over 2 mm size, solution (e.g. water), ratio (e.g. 1:5), and ratio base (e.g. weight /volume). For details see Batjes and van Oostrum (2023). |
| value_avg | Average, for above (it is recommended to use this value for "routine" modelling) |
| dataset_id | Abbreviation for source data set (e.g. WD-ISCN) |
| country_name | Name of country where site is located. |
| longitude | Longitude in degrees (WGS84) |
| latitude | Latitude in degrees (WGS84) |
| positional_uncertainty | Positional uncertainty of the profile's site location (see Table 2). |
| region | Region in which site is located |
| continent | Continent where the profile's site is located |
| date | Date the profile was described/sampled |
| licence | Licence for given data, as indicated by the data provider (i.e. CC BY or CC BY-NC) |

*Format:* All fields in the above files are tab-delimited, with double quotation marks as text delimiters. File coding is according to the UTF-8 unicode transformation format.

*Using the data*: Tutorials for downloading and querying the data, using various platforms, are provided on the WoSIS FAQ webpage (https://www.isric.org/explore/wosis/faq-wosis, last access: 24 April 2024).

**Appendix C: Distribution of sites**

*Insert Table C1 here*

*Insert Table C2 near here*

*Insert Table C3 near here*

**Author contributions.** NB is scientific lead of the WoSIS project and wrote the first draft. LC developed the ETL and GraphQL
procedures while LdS developed the new data model. All authors performed quality checks, data analyses and contributed to the
writing/editing of the final manuscript.

**Competing interests.** The authors declare that they have no conflict of interest.

**Disclaimer**. ISRIC – World Soil Information remains neutral with regard to jurisdictional claims in published maps and
institutional affiliations.

**Acknowledgements**

The development of WoSIS has been made possible thanks to the contributions and shared knowledge of a steadily growing
number of data providers, including soil survey organisations, research institutes and individual experts, for which we are grateful.
Regrettably, we can impossibly acknowledge all contributors (e.g., field surveyors, laboratory personnel, soil experts and database
experts) individually. Therefore, we do this largely in a generic way (see https://www.isric.org/explore/wosis/wosis-contributing-
20 institutions-and-experts; last access: 26 April 2024).

Our special thanks go to Eloi Ribeiro, former WoSIS database management expert at ISRIC, for his sustained support and
advice during the refactoring of the ETL procedure. We also thank our colleague Laura Poggio for useful methodological
discussions concerning methodological linkages between WoSIS and SoilGrids.

We gratefully acknowledge Dr Alessandro Samuel Rosa, the second anonymous reviewer, and the topic editor Dr Sybille
Haßler for their comments which greatly improved the scope of the initial submission.

The ETL procedures and new data model were co-developed in the framework of the European Union's Horizon 2020 HoliSoils project (Grant agreement 101000289) and ISRIC's "global soil information and standards" workstream.

ISRIC − World Soil Information, legally registered as International Soil Reference and Information Centre, receives core funding from the Dutch Government.

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

**Table 1.** Number of soil profiles and properties served in successive WoSIS snapshots.

| Snapshot | No. of profiles | No. of properties[a] |
|---|---|---|
| 2016-07 | 96k | 22 |
| 2019-09 | 196k | 45 |
| 2023-12 | 228k | 45 |

[a] Property names are based on "operational definitions", i.e. a combination of a property and procedure in the terminology of the WoSIS data model (see Sect. 2.4 and Sect. 3.3).

**Table 2**. Positional uncertainty of profile site locations.

| Positional uncertainty | Number of profiles | |
|---|---|---|
| | n | % |
| ~ 100 m | 195,554 | 86 |
| 100 m - 1 km | 21,653 | 9 |
| 1 km – 10 km | 3,846 | 2 |
| Over 10 km | 7,037 | 3 |

**Table 3.** Period of sampling/analysis.

| Period | N of profiles | Percentage |
|---|---|---|
| < 1920 | 37 | 0 |
| 1920-1940 | 253 | 0.1 |
| 1940-1960 | 8,632 | 3.8 |
| 1960-1980 | 35,358 | 15.5 |
| 1980-2000 | 75,686 | 33.2 |
| 2000-2020 | 47,768 | 20.9 |
| Not specified | 60,356 | 26.5 |

**Table 4.** Number of soil profiles per continent.

| Continent | Number of profiles | | |
|---|---|---|---|
| | 2023-snapshot | 2019-snapshot | 2016-snapshot |
| Africa | 32,198 | 27,688 | 17,153 |
| Antarctica | 35 | 9 | 0 |
| Asia | 7,763 | 6,704 | 3,089 |
| Europe | 39,728 | 35,311 | 1,908 |
| North America | 78,996 | 73,604 | 63,066 |
| Oceania | 43,013 | 42,918 | 235 |
| South America | 26,457 | 10,218 | 8,790 |

**Table 5.** Maximum depth of soil sampled per continent.

| Continent | Maximum depth sampled (cm) | | |
|---|---|---|---|
| | <=30 | 30-60 | >60 |
| Africa | 5307 | 1779 | 22458 |
| Antarctica | 6 | 7 | 17 |
| Asia | 635 | 505 | 4310 |
| Europe | 9411 | 2848 | 23195 |
| Northern America | 6190 | 6728 | 60698 |
| Oceania | 9216 | 3792 | 27839 |
| South America | 1730 | 810 | 8477 |

**Table A1.** Coding conventions for observations (i.e. a combination of property, procedure and unit of measurement), number of profiles and layers provided in the WoSIS 2023-snapshot, and inferred accuracy of measurements (Codes are listed in alphabetical order).

| Code | Property | Procedure [a] | Unit | Profiles | Layers | Accuracy (± %) [b] |
|---|---|---|---|---|---|---|
| BDFI33 | Bulk density fine earth [c] | Bulk density of a soil sample that has been desorbed to 33 kPa (1/3 bar) | kg/dm³ | 14886 | 78007 | 25.0 |
| BDFIAD | Bulk density fine earth | Bulk density of a soil sample that has been air dried | kg/dm³ | 4238 | 14485 | 25.0 |
| BDFIFM | Bulk density fine earth | Bulk density of a soil sample at field-soil water content at time of sampling | kg/dm³ | 5265 | 14075 | 25.0 |
| BDFIOD | Bulk density fine earth | Bulk density of a soil sample that has been dried in an oven at 110 °C | kg/dm³ | 26064 | 131623 | 25.0 |
| BDWSAD | Bulk density whole soil [e] | Bulk density of a soil sample that has been air dried | kg/dm³ | 0 | 0 | 25.0 |
| BDWSOD | Bulk density whole soil | Bulk density of a soil sample that has been dried in an oven at 110 °C | kg/dm³ | 14596 | 75397 | 25.0 |
| CECPH7 | Cation exchange capacity | CEC estimated by buffering the soil at "pH7" (e.g., $NH_4Oac$) | cmol(c)/kg | 60339 | 320532 | 20.0 |
| CECPH8 | Cation exchange capacity | CEC estimated by buffering the soil at "pH7" (e.g., $NH_4Oac$) | cmol(c)/kg | 6838 | 25100 | 20.0 |
| CFGR | Coarse fragments | Gravimetric content of soil material larger than 2 mm [c] | g/100g | 39481 | 202414 | 20.0 |
| CFVO | Coarse fragments | Volumetric content of soil material larger than 2 mm [c] | cm³/100cm³ | 48891 | 246580 | 30.0 |
| CLAY | Clay [d] | Determination of total gravimetric content of clay-size fraction (for class-size limits and analytical methods see 'method_options') | g/100g | 153319 | 652347 | 15.0 |
| ECEC | Cation exchange capacity | Effective CEC conventionally approximated by summation of exchangeable bases ($Ca^{2+}$, $Mg^{2+}$, $K^+$, and $Na^+$) plus 1 M KCl exchangeable acidity ($Al^{3+}$ and $H^+$) in acidic soils | cmol(c)/kg | 35123 | 143693 | 25.0 |
| ELCO20 | Electrical conductivity | Electrical conductivity assessed on a 1:2 soil water extract. Used for saline soils. | dS/m | 7971 | 44350 | 10.0 |
| ELCO25 | Electrical conductivity | Electrical conductivity assessed on a 1:2.5 soil water extract. Used for saline soils. | dS/m | 4395 | 17825 | 10.0 |
| ELCO50 | Electrical conductivity | Electrical conductivity assessed on a 1:5 soil water extract. Used for saline soils. | dS/m | 23121 | 90959 | 10.0 |

| ELCOSP | Electrical conductivity | Electrical conductivity assessed on water saturated soil paste. Used for saline soils. | dS/m | 22052 | 85020 | 10.0 |
| NITKJD | Total nitrogen (N) | Kjeldahl wet-oxidation digestion procedure | g/kg | 72905 | 240433 | 10.0 |
| ORGC | Organic carbon (C) | Amount of organic carbon determined according to method specified under 'method_options' | g/kg | 135655 | 526953 | 15.0 |
| ORGM | Organic matter | Determination of organic compounds that accompany soil particles through a 2-mm sieve using loss-on-ignition (LOI) at about 400 degrees Celsius. | g/kg | 3871 | 16282 | 15.0 |
| PHAQ | pH | A measure of the acidity or alkalinity in soils, defined as the negative logarithm (base 10) of the activity of hydronium ions ($H^+$) in water[a]. | unitless | 140326 | 655336 | 0.3 |
| PHCA | pH | A measure of the acidity or alkalinity in soils, defined as the negative logarithm (base 10) of the activity of hydronium ions ($H^+$), in the specified $CaCl_2$ solution. | unitless | 69437 | 325153 | 0.3 |
| PHKC | pH | A measure of the acidity or alkalinity in soils, defined as the negative logarithm (base 10) of the activity of hydronium ions ($H^+$), in the specified KCl solution. | unitless | 38022 | 173464 | 0.3 |
| PHNF | pH | A measure of the acidity or alkalinity in soils, defined as the negative logarithm (base 10) of the activity of hydronium ions ($H^+$), in the specified NaF solution. | unitless | 4965 | 25409 | 0.3 |
| PHETB1 | Phosphorus (P) | Phosphorus determined according to the Bray-I method, a combination of HCl and $NH_4$-F to remove easily acid soluble P forms, largely Al- and Fe-phosphates (mainly applicable for acid soils) | mg/kg | 10719 | 40379 | 40.0 |
| PHETM3 | Phosphorus (P) | Determined according to Mehlich-3 method, a weak acid soil extraction procedure that is considered suitable for removing P and other elements in acid and neutral soil. The extract is composed of 0.2 M glacial acetic acid, 0.25 M ammonium nitrate, 0.015 M ammonium fluoride, 0.013 M nitric acid, and 0.001 M ethylene diamine tetraacetic acid (EDTA). | mg/kg | 1444 | 7230 | 25.0 |
| PHETOL | Phosphorus (P) | Phosphorus determined according to the Olsen method (0.5 M sodium bicarbonate ($NaHCO_3$) solution at a pH of 8.5); used extract P from calcareous, alkaline, and neutral soils. | mg/kg | 4266 | 12291 | 25.0 |
| PHPRTN | Phosphorus (P) | Phosphorus retention measured according to the New Zealand method (Blakemore, 1981). | g/100g | 5599 | 26569 | 20.0 |
| PHPTOT | Phosphorus (P) | Phosphorus determined with a "harsh" digest procedure to liberate and measure all forms of element. | mg/kg | 7561 | 19310 | 15.0 |
| PHPWSL | Phosphorus (P) | Phosphorus soluble in soluble in water | mg/kg | 282 | 1241 | 15.0 |

| | | | | | | |
|---|---|---|---|---|---|---|
| SAND | Sand | Determination of total gravimetric content of sand-size fraction (for class-size limits and analytical methods see 'method_options'). | g/100g | 119127 | 542463 | 15.0 |
| SILT | Silt [f] | Determination of total gravimetric content of silt-size fraction (for class-size limits and analytical methods see 'method_options'). | g/100g | 145906 | 620790 | 15.0 |
| TCEQ | Calcium carbonate equivalent (TCEQ) | Determination of the gravimetric loss of carbonates as carbon dioxide in the presence of excess hydrochloric acid. The quantity of carbonate ($CO_3$) in the soil is expressed as $CaCO_3$ and as a weight percentage of the less than 2 mm size fraction. | g/kg | 59294 | 247368 | 10.0 |
| TOTC | Total carbon (C) | Total C is quantified by two basic methods: wet or dry combustion (see 'method options'). In total C determinations, all forms of C in a soil are converted to $CO_2$ followed by a quantification of the evolved $CO_2$. Total C can be used to estimate the organic C content of a soil. The difference between total and inorganic C is an estimate of the organic C. | g/kg | 33527 | 112787 | 10.0 |
| WG0006 | Water retention gravimetric | Water retention assessed at tension 6 kPa (see method options') | g/100g | 827 | 3828 | 20.0 |
| WG0010 | Water retention gravimetric | Water retention assessed at tension 10 kPa (see 'method_options'). | g/100g | 2970 | 12517 | 20.0 |
| WG0033 | Water retention gravimetric | Water retention assessed at tension 33 kPa (see 'method_options'). | g/100g | 20994 | 94707 | 20.0 |
| WG0100 | Water retention gravimetric | Water retention assessed at tension 100 kPa (see 'method_options'). | g/100g | 687 | 3360 | 20.0 |
| WG0200 | Water retention gravimetric | Water retention assessed at tension 200 kPa (see 'method_options'). | g/100g | 4391 | 27773 | 20.0 |
| WG0500 | Water retention gravimetric | Water retention assessed at tension 500 kPa (see 'method_options'). | g/100g | 326 | 1414 | 20.0 |
| WG1500 | Water retention gravimetric | Water retention assessed at tension 1500 kPa (see 'method_options'). | g/100g | 33782 | 181999 | 20.0 |
| WV0010 | Water retention volumetric | Water retention assessed at tension 10 kPa (see 'method_options'). | cm³/100cm³ | 1914 | 6883 | 20.0 |
| WV0033 | Water retention volumetric | Water retention assessed at tension 33 kPa (see 'method_options'). | cm³/100cm³ | 7444 | 22291 | 20.0 |
| WV0100 | Water retention volumetric | Water retention assessed at tension 100 kPa (see 'method_options'). | cm³/100cm³ | 747 | 2553 | 20.0 |
| WV0500 | Water retention volumetric | Water retention assessed at tension 500 kPa (see 'method_options'). | cm³/100cm³ | 702 | 1758 | 20.0 |
| WV1500 | Water retention volumetric | Water retention assessed at tension 1500 kPa (see 'method_options'). | cm³/100cm³ | 7904 | 23331 | 20.0 |

[a] Method options for each analytical procedure are described in Batjes and van Oostrum (2023), and provided in file *Wosis_202312_xxxx.tsv,* see Appendix C.

[b] Inferred accuracy (or uncertainty), rounded to the nearest 5%, unless otherwise indicated (i.e. units for soil pH) as derived from the following sources (Al-Shammary et al., 2018; Kalra and Maynard, 1991; Rayment and Lyons, 2011; Rossel and McBratney, 1998; van Reeuwijk, 1983; WEPAL, 2019). These figures are first approximations that should be fine-tuned once more specific results of laboratory proficiency tests, resp. national Soil Quality Management systems, become freely available (e.g. from the GLOSOLAN laboratory proficiency programme).

[c] Generally, the fine earth fraction is defined as being < 2 mm. Alternatively, an upper limit of 1 mm was used in the former Soviet Union and its satellite states (Katchynsky scheme). The actual size limits are specified under "method options" (see Appendix C).

[d] Provided only when the sum of clay, silt and sand fraction is $\geq$ 90 and $\leq$ 100 percent (Note that users should normalise the totals to 100 percent before using them for mapping or modelling purposes; further, more stringent limits (e.g. $\geq$ 98 and $\leq$ 102) may be considered).

[e] No data are being served for this property because the associated licences are flagged as 'restricted' by the data providers.

[f] The lower and upper limits for the 'silt' size fraction can vary markedly between countries, hence these limits have been specified explicitly in WoSIS under "method options" (see Appendix B). Development and application of conversion procedures to one common "silt" fraction (e.g. 0.002-0.05 mm) is beyond the remit of the WoSIS project itself. The necessary pedotransfer functions should be developed (and tested) prior to generating particle size class related soil property maps for a given geography. Research in this direction is being undertaken by the SoilGrids team, based on the "best available" comparative datasets for calibration.

**Table A2.** Coding conventions and brief descriptions for soil classification, horizon designations and number of occurrences in the WoSIS 2023-snapshot.

| Code | Description | Count |
|------|-------------|-------|
| CSTX | Classification of the soil profile according to specified edition (year) of USDA Soil Taxonomy, at least at soil order level | 31400 |
| CWRB | Classification of the soil profile according to specified edition (year) of the World Reference Base for Soil Resources (WRB), at least at reference soil group level | 39649 |
| CFAO | Classification of the soil profile according to specified edition (year) of the FAO-Unesco Legend, at least at major group level | 38792 |
| HODS[a] | Horizon designations as provided in the source databases | 80849 / 396522[b] |

[a] Where available, the "cleaned" (original) layer/horizon designation is provided for general information; these codes have not been standardised as they vary widely between different classification systems (Bridges, 1993; Gerasimova et al., 2013). When no horizon designations are provided in the source data bases, we have flagged all layers with an upper depth given as being negative (e.g. -10 to 0 cm that is using pre-1993 conventions (see Sect. 3.1) in the source databases as likely being a shallow "organic surface" layer above a mineral soil layer.

[b] Number of profiles with horizon descriptions respectively total number of layers with horizon designations.

**Table C1.** Number of sites per continent and country.

| Continent | Country | Country code | No. of sites | Area (km²) | Site density (per 1000 km²) |
|---|---|---|---|---|---|
| Africa | Abyei | 4 | 0 | 9943 | 0 |
| | Algeria | DZ | 10 | 2308647 | 0.004 |
| | Angola | AO | 1168 | 1246690 | 0.937 |
| | Benin | BJ | 743 | 115247 | 6.447 |
| | Botswana | BW | 994 | 578247 | 1.719 |
| | British Indian Ocean Territory | IO | 0 | 49 | 0 |
| | Burkina Faso | BF | 2023 | 273281 | 7.403 |
| | Burundi | BI | 36 | 26857 | 1.34 |
| | Cameroon | CM | 1417 | 465363 | 3.045 |
| | Cape Verde | CV | 0 | 4056 | 0 |
| | Central African Republic | CF | 88 | 619591 | 0.142 |
| | Chad | TD | 7 | 1265392 | 0.006 |
| | Comoros | KM | 0 | 1652 | 0 |
| | Congo | CG | 70 | 340599 | 0.206 |
| | Côte d'Ivoire | CI | 255 | 321762 | 0.793 |
| | Democratic Republic of the Congo | CD | 378 | 2329162 | 0.162 |
| | Djibouti | DJ | 0 | 21670 | 0 |
| | Egypt | EG | 26 | 982161 | 0.026 |
| | Equatorial Guinea | GQ | 0 | 27000 | 0 |
| | Eritrea | ER | 0 | 120763 | 0 |
| | Ethiopia | ET | 1712 | 1129314 | 1.516 |
| | Gabon | GA | 47 | 264022 | 0.178 |
| | Gambia | GM | 0 | 11203 | 0 |
| | Ghana | GH | 432 | 238842 | 1.809 |
| | Guinea | GN | 128 | 243023 | 0.527 |
| | Guinea-Bissau | GW | 15 | 30740 | 0.488 |
| | Hala'ib triangle | 10 | 0 | 17684 | 0 |

| | | | | | |
|---|---|---|---|---|---|
| Ilemi triangle | | 13 | 0 | 3179 | 0 |
| Kenya | KE | | 1603 | 582342 | 2.753 |
| Lesotho | LS | | 33 | 30453 | 1.084 |
| Liberia | LR | | 50 | 96103 | 0.52 |
| Libya | LY | | 14 | 1620583 | 0.009 |
| Madagascar | MG | | 130 | 588834 | 0.221 |
| Malawi | MW | | 3050 | 118715 | 25.692 |
| Mali | ML | | 885 | 1251471 | 0.707 |
| Ma'tan al-Sarra | | 11 | 0 | 1993 | 0 |
| Mauritania | MR | | 13 | 1038527 | 0.013 |
| Mauritius | MU | | 0 | 2014 | 0 |
| Mayotte | YT | | 0 | 378 | 0 |
| Morocco | MA | | 113 | 414030 | 0.273 |
| Mozambique | MZ | | 565 | 787305 | 0.718 |
| Namibia | NA | | 1569 | 823989 | 1.904 |
| Niger | NE | | 520 | 1182602 | 0.44 |
| Nigeria | NG | | 1402 | 908978 | 1.542 |
| Réunion | RE | | 0 | 2504 | 0 |
| Rwanda | RW | | 1016 | 25388 | 40.018 |
| Saint Helena, Ascension and Tristan da Cunha | SH | | 0 | 399 | 0 |
| Sao Tome and Principe | ST | | 0 | 991 | 0 |
| Senegal | SN | | 312 | 196200 | 1.59 |
| Seychelles | SC | | 0 | 499 | 0 |
| Sierra Leone | SL | | 12 | 72281 | 0.166 |
| Somalia | SO | | 245 | 632562 | 0.387 |
| South Africa | ZA | | 879 | 1220127 | 0.72 |
| South Sudan | SS | | 82 | 629821 | 0.13 |
| Sudan | SD | | 130 | 1843196 | 0.071 |
| Swaziland | SZ | | 14 | 17290 | 0.81 |
| Togo | TG | | 9 | 56767 | 0.159 |
| Tunisia | TN | | 60 | 155148 | 0.387 |
| Uganda | UG | | 84 | 241495 | 0.348 |
| United Republic of Tanzania | TZ | | 1910 | 939588 | 2.033 |

| | | | | | | |
|---|---|---|---|---|---|---|
| | Western Sahara | EH | | 0 | 268617 | 0 |
| | Zambia | ZM | | 603 | 751063 | 0.803 |
| | Zimbabwe | ZW | | 413 | 390648 | 1.057 |
| | | | | | | |
| Antarctica | Antarctica | AQ | | 30 | 12537967 | 0.002 |
| | Bouvet Island | BV | | 0 | 45 | 0 |
| | French Southern and Antarctic Territories | TF | | 0 | 7738 | 0 |
| | Heard Island and McDonald Islands | HM | | 0 | 412 | 0 |
| | South Georgia and the South Sandwich Islands | GS | | 0 | 3870 | 0 |
| | | | | | | |
| Asia | Afghanistan | AF | | 19 | 641827 | 0.03 |
| | Aksai Chin | | 1 | 0 | 30666 | 0 |
| | Armenia | AM | | 509 | 29624 | 17.182 |
| | Arunachal Pradesh | | 2 | 2 | 67965 | 0.029 |
| | Azerbaijan | AZ | | 28 | 164780 | 0.17 |
| | Bahrain | BH | | 2 | 673 | 2.97 |
| | Bangladesh | BD | | 207 | 139825 | 1.48 |
| | Bhutan | BT | | 85 | 37674 | 2.256 |
| | Brunei Darussalam | BN | | 0 | 5899 | 0 |
| | Cambodia | KH | | 424 | 181424 | 2.337 |
| | China | CN | | 1644 | 9345214 | 0.176 |
| | China/India | | 3 | 0 | 3526 | 0 |
| | Christmas Island | CX | | 0 | 136 | 0 |
| | Cocos (Keeling) Islands | CC | | 0 | 16 | 0 |
| | Cyprus | CY | | 12 | 9249 | 1.297 |
| | Democratic People's Republic of Korea | KP | | 0 | 122465 | 0 |
| | Georgia | GE | | 18 | 69785 | 0.258 |
| | Hong Kong | HK | | 2 | 1081 | 1.851 |
| | India | IN | | 199 | 2961118 | 0.067 |
| | Indonesia | ID | | 179 | 1888620 | 0.095 |
| | Iran (Islamic Republic of) | IR | | 2010 | 1677319 | 1.198 |

| | | | | | |
|---|---|---|---|---|---|
| Iraq | IQ | | 14 | 435864 | 0.032 |
| Israel | IL | | 17 | 20720 | 0.82 |
| Jammu and Kashmir | | 12 | 4 | 186035 | 0.022 |
| Japan | JP | | 197 | 373651 | 0.527 |
| Jordan | JO | | 47 | 89063 | 0.528 |
| Kazakhstan | KZ | | 52 | 2841103 | 0.018 |
| Kuril islands | | 5 | 0 | 4996 | 0 |
| Kuwait | KW | | 1 | 17392 | 0.057 |
| Kyrgyzstan | KG | | 1 | 199188 | 0.005 |
| Lao People's Democratic Republic | LA | | 20 | 230380 | 0.087 |
| Lebanon | LB | | 10 | 10136 | 0.987 |
| Macau | MO | | 0 | 17 | 0 |
| Malaysia | MY | | 155 | 329775 | 0.47 |
| Maldives | MV | | 0 | 223 | 0 |
| Mongolia | MN | | 9 | 1564529 | 0.006 |
| Myanmar | MM | | 0 | 667085 | 0 |
| Nepal | NP | | 142 | 147437 | 0.963 |
| Occupied Palestinian Territory | PS | | 18 | 6225 | 2.892 |
| Oman | OM | | 11 | 308335 | 0.036 |
| Pakistan | PK | | 45 | 788439 | 0.057 |
| Paracel Islands | | 6 | 0 | 8 | 0 |
| Philippines | PH | | 78 | 296031 | 0.263 |
| Qatar | QA | | 0 | 11549 | 0 |
| Republic of Korea | KR | | 23 | 99124 | 0.232 |
| Saudi Arabia | SA | | 7 | 1925621 | 0.004 |
| Scarborough Reef | | 7 | 0 | 44 | 0 |
| Senkaku Islands | | 8 | 0 | 5 | 0 |
| Singapore | SG | | 1 | 594 | 1.683 |
| Spratly Islands | | 9 | 0 | 1 | 0 |
| Sri Lanka | LK | | 73 | 66173 | 1.103 |
| Syrian Arab Republic | SY | | 69 | 188128 | 0.367 |
| Taiwan | TW | | 35 | 36127 | 0.969 |
| Tajikistan | TJ | | 5 | 142004 | 0.035 |
| Thailand | TH | | 479 | 515417 | 0.929 |

| | | | | | |
|---|---|---|---|---|---|
| | Timor-Leste | TL | 0 | 14892 | 0 |
| | Turkey | TR | 69 | 781229 | 0.088 |
| | Turkmenistan | TM | 0 | 555052 | 0 |
| | United Arab Emirates | AE | 12 | 71079 | 0.169 |
| | Uzbekistan | UZ | 9 | 449620 | 0.02 |
| | Viet Nam | VN | 29 | 327575 | 0.089 |
| | Yemen | YE | 284 | 453596 | 0.626 |
| | | | | | |
| Europe | Albania | AL | 97 | 28682 | 3.382 |
| | Andorra | AD | 0 | 475 | 0 |
| | Austria | AT | 128 | 83964 | 1.524 |
| | Belarus | BY | 96 | 207581 | 0.462 |
| | Belgium | BE | 7013 | 30669 | 228.667 |
| | Bosnia and Herzegovina | BA | 32 | 51145 | 0.626 |
| | Bulgaria | BG | 134 | 111300 | 1.204 |
| | Croatia | HR | 78 | 56589 | 1.378 |
| | Czech Republic | CZ | 666 | 78845 | 8.447 |
| | Denmark | DK | 72 | 44458 | 1.619 |
| | Estonia | EE | 241 | 45441 | 5.304 |
| | Faroe Islands | FO | 0 | 1400 | 0 |
| | Finland | FI | 442 | 336892 | 1.312 |
| | France | FR | 3183 | 548785 | 5.8 |
| | Germany | DE | 4362 | 357227 | 12.211 |
| | Gibraltar | GI | 0 | 6 | 0 |
| | Greece | GR | 374 | 132549 | 2.822 |
| | Guernsey | GG | 0 | 79 | 0 |
| | Holy See | VA | 0 | 0 | 0 |
| | Hungary | HU | 1421 | 93119 | 15.26 |
| | Iceland | IS | 17 | 102566 | 0.166 |
| | Ireland | IE | 124 | 69809 | 1.776 |
| | Isle of Man | IM | 0 | 573 | 0 |
| | Italy | IT | 576 | 301651 | 1.909 |
| | Jersey | JE | 0 | 120 | 0 |
| | Latvia | LV | 102 | 64563 | 1.58 |

| | | | | |
|---|---|---|---|---|
| | Liechtenstein | LI | 0 | 151 | 0 |
| | Lithuania | LT | 127 | 64943 | 1.956 |
| | Luxembourg | LU | 142 | 2621 | 54.184 |
| | Malta | MT | 0 | 316 | 0 |
| | Monaco | MC | 0 | 8 | 0 |
| | Montenegro | ME | 12 | 13776 | 0.871 |
| | Netherlands | NL | 958 | 35203 | 27.214 |
| | Norway | NO | 507 | 324257 | 1.564 |
| | Poland | PL | 796 | 311961 | 2.552 |
| | Portugal | PT | 455 | 91876 | 4.952 |
| | Republic of Moldova | MD | 35 | 33798 | 1.036 |
| | Romania | RO | 113 | 238118 | 0.475 |
| | Russian Federation | RU | 1464 | 16998830 | 0.086 |
| | San Marino | SM | 0 | 60 | 0 |
| | Serbia | RS | 69 | 88478 | 0.78 |
| | Slovakia | SK | 161 | 49072 | 3.281 |
| | Slovenia | SI | 67 | 20320 | 3.297 |
| | Spain | ES | 907 | 505752 | 1.793 |
| | Svalbard and Jan Mayen Islands | SJ | 4 | 63464 | 0.063 |
| | Sweden | SE | 594 | 449212 | 1.322 |
| | Switzerland | CH | 10928 | 41257 | 264.874 |
| | The Republic of North Macedonia | MK | 20 | 25424 | 0.787 |
| | Ukraine | UA | 462 | 600526 | 0.769 |
| | United Kingdom | GB | 1727 | 244308 | 7.069 |
| | | | | | |
| Northern America | Anguilla | AI | 0 | 79 | 0 |
| | Antigua and Barbuda | AG | 0 | 452 | 0 |
| | Aruba | AW | 0 | 180 | 0 |
| | Bahamas | BS | 0 | 11904 | 0 |
| | Barbados | BB | 3 | 433 | 6.928 |
| | Belize | BZ | 26 | 21764 | 1.195 |
| | Bermuda | BM | 0 | 63 | 0 |
| | British Virgin Islands | VG | 0 | 154 | 0 |
| | Canada | CA | 8778 | 9875646 | 0.889 |

| | | | | |
|---|---|---|---|---|
| | Cayman Islands | KY | 0 | 269 | 0 |
| | Clipperton Island | CP | 0 | 9 | 0 |
| | Costa Rica | CR | 560 | 51042 | 10.971 |
| | Cuba | CU | 53 | 110863 | 0.478 |
| | Dominica | DM | 0 | 751 | 0 |
| | Dominican Republic | DO | 10 | 48099 | 0.208 |
| | El Salvador | SV | 38 | 20732 | 1.833 |
| | Greenland | GL | 2 | 2165159 | 0.001 |
| | Grenada | GD | 0 | 318 | 0 |
| | Guadeloupe | GP | 5 | 1697 | 2.947 |
| | Guatemala | GT | 28 | 109062 | 0.257 |
| | Haiti | HT | 0 | 27022 | 0 |
| | Honduras | HN | 38 | 112124 | 0.339 |
| | Jamaica | JM | 74 | 10965 | 6.749 |
| | Martinique | MQ | 0 | 1104 | 0 |
| | Mexico | MX | 12599 | 1949527 | 6.463 |
| | Montserrat | MS | 0 | 101 | 0 |
| | Netherlands Antilles | AN | 4 | 790 | 5.066 |
| | Nicaragua | NI | 21 | 128376 | 0.164 |
| | Panama | PA | 50 | 74850 | 0.668 |
| | Puerto Rico | PR | 280 | 8937 | 31.329 |
| | Saint Kitts and Nevis | KN | 0 | 262 | 0 |
| | Saint Lucia | LC | 0 | 603 | 0 |
| | Saint Pierre and Miquelon | PM | 0 | 233 | 0 |
| | Saint Vincent and the Grenadines | VC | 0 | 427 | 0 |
| | Trinidad and Tobago | TT | 2 | 5144 | 0.389 |
| | Turks and Caicos Islands | TC | 0 | 530 | 0 |
| | United States Minor Outlying Islands | UM | 0 | 348 | 0 |
| | United States of America | US | 56322 | 9315946 | 6.046 |
| | United States Virgin Islands | VI | 46 | 352 | 130.555 |
| | | | | | |
| Oceania | American Samoa | AS | 0 | 200 | 0 |
| | Australia | AU | 42767 | 7687634 | 5.563 |

| | | | | | |
|---|---|---|---|---|---|
| | Cook Islands | CK | 0 | 241 | 0 |
| | Fiji | FJ | 6 | 18293 | 0.328 |
| | French Polynesia | PF | 0 | 3967 | 0 |
| | Guam | GU | 15 | 544 | 27.579 |
| | Kiribati | KI | 0 | 1020 | 0 |
| | Marshall Islands | MH | 0 | 268 | 0 |
| | Micronesia (Federated States of) | FM | 75 | 740 | 101.343 |
| | Nauru | NR | 0 | 22 | 0 |
| | New Caledonia | NC | 2 | 18574 | 0.108 |
| | New Zealand | NZ | 52 | 270415 | 0.192 |
| | Niue | NU | 0 | 263 | 0 |
| | Norfolk Island | NF | 0 | 38 | 0 |
| | Northern Mariana Islands | MP | 0 | 476 | 0 |
| | Palau | PW | 18 | 451 | 39.924 |
| | Papua New Guinea | PG | 24 | 462230 | 0.052 |
| | Pitcairn Islands | PN | 0 | 49 | 0 |
| | Samoa | WS | 18 | 2835 | 6.349 |
| | Solomon Islands | SB | 1 | 28264 | 0.035 |
| | Tokelau | TK | 0 | 15 | 0 |
| | Tonga | TO | 0 | 700 | 0 |
| | Tuvalu | TV | 0 | 48 | 0 |
| | Vanuatu | VU | 1 | 12236 | 0.082 |
| | Wake Island | WK | 0 | 7 | 0 |
| | Wallis and Futuna Islands | WF | 0 | 142 | 0 |
| South America | Argentina | AR | 253 | 2780175 | 0.091 |
| | Bolivia (Plurinational State of) | BO | 87 | 1084491 | 0.08 |
| | Brazil | BR | 9262 | 8485946 | 1.091 |
| | Chile | CL | 13662 | 753355 | 18.135 |
| | Colombia | CO | 236 | 1137939 | 0.207 |
| | Ecuador | EC | 94 | 256249 | 0.367 |
| | Falkland Islands (Malvinas) | FK | 0 | 12084 | 0 |
| | French Guiana | GF | 30 | 83295 | 0.36 |
| | Guyana | GY | 43 | 211722 | 0.203 |
| | Paraguay | PY | 2 | 399349 | 0.005 |

| | | | | |
|---|---|---:|---:|---:|
| Peru | PE | 158 | 1290640 | 0.122 |
| Suriname | SR | 31 | 145100 | 0.214 |
| Uruguay | UY | 136 | 177811 | 0.765 |
| Venezuela (Bolivarian Republic of) | VE | 204 | 912025 | 0.224 |

[a] Disputed territory. Country names and areas are based on the Global Administrative Layers (GAUL) database, see: https://data.apps.fao.org/map/catalogsrv/eng/catalog.search?id=12691#/metadata/9c35ba10-5649-41c8-bdfc-eb78e9e65654 (last access: 26 April 2024).

**Table C2.** Number of sites by world terrestrial ecosystems (WTE)[a].

| Temperature zone | Moisture zone | No. of sites | Percent (%) |
|---|---|---:|---:|
| Polar | Dry | 224 | 0.1 |
| Polar | Moist | 532 | 0.2 |
| Boreal | Dry | 1789 | 0.8 |
| Boreal | Moist | 3398 | 1.6 |
| Cool Temperate | Desert | 25 | 0 |
| Cool Temperate | Dry | 10968 | 5 |
| Cool Temperate | Moist | 53245 | 24.5 |
| Warm Temperate | Desert | 238 | 0.1 |
| Warm Temperate | Dry | 29209 | 13.4 |
| Warm Temperate | Moist | 46533 | 21.4 |
| Sub Tropical | Desert | 296 | 0.1 |
| Sub Tropical | Dry | 25748 | 11.8 |
| Sub Tropical | Moist | 17906 | 8.2 |
| Tropical | Desert | 178 | 0.1 |
| Tropical | Dry | 11315 | 5.2 |
| Tropical | Moist | 11095 | 5.1 |
| No data | - | 4674 | 2.2 |

[a] World Terrestrial Ecosystems (WTE) as defined by Sayre (2022). Total may differ from 100% due to rounding

Table C3. Number of sites by WWF biome[b].

| WWF biome | No. of sites | Percent (%) |
|---|---|---|
| Boreal Forests/Taiga | 5519 | 2.5 |
| Deserts and Xeric Shrublands | 13410 | 6.2 |
| Flooded Grasslands and Savannas | 792 | 0.4 |
| Lakes | 85 | 0 |
| Mangroves | 765 | 0.4 |
| Mediterranean Forests, Woodlands, and Scrub | 24459 | 11.3 |
| Montane Grasslands and Shrublands | 2796 | 1.3 |
| Rock and Ice | 20 | 0 |
| Temperate Broadleaf and Mixed Forests | 74068 | 34.1 |
| Temperate Coniferous Forests | 14436 | 6.6 |
| Temperate Grasslands, Savannas, and Shrublands | 23890 | 11 |
| Tropical and Subtropical Coniferous Forests | 2363 | 1.1 |
| Tropical and Subtropical Dry Broadleaf Forests | 4120 | 1.9 |
| Tropical and subtropical grasslands, savannas, and shrublands | 31376 | 14.4 |
| Tropical and Subtropical Moist Broadleaf Forests | 16478 | 7.6 |
| Tundra | 2072 | 1 |
| No data | 724 | 0.3 |

[a] Biomes defined according to "Terrestrial Ecoregions of the World" (WWF) (Olson et al., 2001a). Total may differ from 100% due to rounding.

**Figure captions**

**Figure 1.** Schematic WoSIS workflow for ingesting, standardising and distributing soil profile data.

5    **Figure 2.** Distribution of sites represented in the 2023 snapshot of WoSIS (Good homolosine equal-area projection).

**Figure 3**. Density and spatial distribution of profiles served with the 2017, 2019 and 2023 WoSIS snapshots.