# Peer review of "Providing quality-assessed and standardised soil data to support global mapping and modelling (WoSIS snapshot 2023)"

_Earth System Science Data, 2024_

## Author Comment (AC1)

**Response to Reviewer 1 (**essd-2024-14-RC1)

Please find below our response to your insightful comments on our manuscript; thanks for this.

Our responses are shown in blue respectively black font for textual changes to the manuscript (Times Roman 10).

We will finalise the revised manuscript once we have received the comments from the second reviewer. At that stage, textual changes will be visible, i.e. flagged using "track changes", in a separate copy of the revised paper.

**GENERAL COMMENTS**

ISRIC World Soil Information has long been recognized for its efforts in collecting, organizing, and disseminating quality soil data for the international community. The current paper, the third in the series, succinctly describes the procedures adopted by ISRIC for cleaning and disseminating global soil profile data. The international community will certainly appreciate the release of the third static WoSIS snapshot, providing quality-assessed and standardized data for 228k geo-referenced profiles. Similar to the previous two manuscripts, the present manuscript aims to inform soil data producers and users about the current status of WoSIS and the availability of standardized point soil data for digital soil mapping and earth system modeling. Overall, I believe that the manuscript delivers this information in a clear, concise, and organized manner.

We thank the reviewer for his positive comments and kind recognition of ISRIC's long standing efforts in collecting, standardising and serving quality-assessed soil data to the international community, as far as permitted by the licences indicated by the data providers.

The present manuscript is the third in a series of manuscripts describing WoSIS snapshots. As ISRIC continues its role in collating more soil data from multiple sources, we expect to see new snapshots and accompanying manuscripts in the coming years. For this reason, I believe that the present manuscript should be prepared in a manner that presents the history of WoSIS and how its soil data processing strategies have evolved over time. The current paper reads more like a report and provides insufficient context regarding its relation to the previous two versions. While informative for someone familiar with WoSIS, it does not adequately educate new users about the database's past history. Adding more historical context to the manuscript would greatly enhance its comprehensiveness and usefulness.

Thanks. We have added a short "historical perspective" to the manuscript, as recommended, showing main developments since the conceptualisation of WoSIS in 2009. The following text has been added to the introduction, as "second" paragraph.

Prior to describing the "2023 snapshot" itself, we first provide a short retrospective on WoSIS developments. In the early days of desktop computers, ISRIC with its partners compiled a range of project-specific databases, such as ISIS (van de Ven and Tempel 1994), created to manage data for the ISRIC World Soil Reference Collection, several national and continental scale Soil and Terrain Databases (SOTER, e.g.,FAO 1989; Oldeman and van Engelen 1993; FAO *et al.* 1998; FAO *et al.* 2007), the WISE database (Batjes and Bridges 1994; Batjes *et al.* 1995), and the Africa Soil Profile Database (Leenaars *et al.* 2014). While most of these databases were structured along the general principles and criteria of the FAO Guidelines for soil description (FAO 1990, 2006) and USDA Soil Survey Manual (Soil Survey Division Staff 1993), the ISIS, SOTER, WISE, and AFSP databases each had their own data models and conventions. Further, the databases were implemented on stand-alone computers, using a range of commercial software products. In 2009 ISRIC decided to bring the above stand-alone products together in a centralised enterprise database, known as WoSIS (World Soil Information Service), developed using PostgreSQL with the PostGIS extension for handling spatial data. After the initial ingest and standardisation of the above 'ISRIC holdings', the service was to be expanded with datasets shared by a diverse range of soil data providers.

The original aim of WoSIS was to accommodate any type of soil data (profile, vector and grid) (Tempel *et al.* 2013; Ribeiro *et al.* 2015). However, from 2015 onwards, in view of technical considerations and institutional developments, the scope of WoSIS changed to "safeguarding, processing, standardising and serving geo-referenced soil profile (point) data for the world (Ribeiro *et al.* 2020). Alternatively, vector and grid data/maps derived from traditional soil mapping (e.g., Dijkshoorn *et al.* 2005; van Engelen *et al.* 2006; FAO *et al.* 2012; Batjes 2016) and digital soil mapping (Hengl *et al.* 2017; Poggio *et al.* 2021; Turek *et al.* 2022) are managed and served through other components of our spatial data infrastructure, such as the ISRIC data hub (https://data.isric.org) and the SoilGrids/WoSIS portal (https://soilgrids.org; last access: 24 April 2024). All these services are developed using free-and-open-software (FOSS).

In view of the above changes we have moved part of the original introduction to Sect. 2.4, renamed as "Standardising soil property names". Further, we added the following text to explain the rationale, or need, for implementing several methodological changes to the WoSIS data model and workflow (see Section 2.1, second sentence):

This proved necessary as this procedure was essentially designed as a series of dataset-specific python and SQL scripts, which was adequate as long as WoSIS was still relatively small. However, in view of the rapidly growing population of "shared" soil data and overall complexity of the data model itself (Ribeiro *et al.* 2020) it proved necessary to implement a new, state-of-the-art, ISO domain model (de Sousa 2023; de Sousa *et al.* 2023), with refactored ETL (extract, transform and load) procedures, to ultimately better serve our diverse user community in our capacity as World Data Centre for Soils (WDC-Soils).

The manuscript could benefit from further discussion on the future trajectory of WoSIS. The initial snapshot in 2016 contained 96K soil profiles, with a significant increase to 196k profiles in the second snapshot in 2019. However, the third snapshot in 2023, four years later, only brought the count to 228k profiles, accompanied by a notable rise in restricted soil profiles within WoSIS's internal database. Given the finite nature of soil data, I anticipate that future snapshots may exhibit smaller increases in data availability. This trend mirrors our experiences in Brazil with efforts to rescue legacy soil data for

inclusion in the Brazilian Soil Data Repository (Soil Data). I am curious to know if you share this expectation and whether the manuscript will address strategies already in place or those to be implemented to continue enhancing soil data availability.

Thanks for this comment! This is indeed a source of concern for the future development of open source databases for the benefit of the international community. We also noted that getting positive responses to our request for sharing soil data for consideration in WoSIS becomes increasingly "difficult", and time demanding (success rate less than 15%). Further, increasingly many soil datasets are shared with the provision that coordinates may not be shown and hence cannot be served from WoSIS as snapshots (and *wosis_latest*). Another source of concern is that many new soil data collection activities, such as soil monitoring programmes (e.g., Ballabio *et al.* 2016; Orgiazzi *et al.* 2018), focus only on the top 20 or 30 cm of the soil, that is they do not consider the full soil profile depth as required for comprehensive soil assessments (e.g.,Leenaars *et al.* 2018; von Haden *et al.* 2020; Wang *et al.* 2022).

The above elements/considerations have been added in the revised manuscript (please see annotated version of revision), see Section 6. Discussion, page 16, 2nd and 3rd paragraph:

It is our intention to gradually fill gaps in the geographic distribution (Fig. 3) and range of soil properties (Appendix A) in the coming years. This work is part of ISRIC's remit as a regular member of the World Data System (https://worlddatasystem.org). The degree to which this will be feasible however, will largely depend on the willingness and ability of data providers to share (some of) their data for consideration in WoSIS. For the northern Boreal and Arctic region, for example, ISRIC can draw on new profiles collated by the International Soil Carbon Network (ISCN, see Malhotra *et al.* 2019). Alternatively, it should be reiterated that some datasets in our repository (e.g. ICP Forests 2021) can *only* be standardised and used for SoilGrids™ applications by ISRIC due to existing licence restrictions.

Concerning the scope for expanding *wosis_latest* in the coming years, we noted that getting positive responses to our requests for sharing soil data is becoming increasingly "difficult"; the overall success rate is around 15%. Further, increasingly soil datasets are now being shared with the provision that profile coordinates may not be shown and hence cannot be served to our user community through *wosis_latest*. Another source of concern is that many new soil data collection activities, such as soil monitoring programmes (e.g., Ballabio *et al.* 2016; Orgiazzi *et al.* 2018), focus only on the top 20 or 30 cm of the soil, that is do not consider the actual soil profile depth as required for comprehensive soil assessments (e.g.,Batlle-Bayer *et al.* 2010; Leenaars *et al.* 2018; von Haden *et al.* 2020; Wang *et al.* 2022).

I would like to offer specific feedback on three sections of the manuscript, which align with my earlier observations on the manuscript's overall quality. I trust that you will find my comments valuable.

**SPECIFIC COMMENTS**

**WoSIS data model and workflow**

While I appreciate the explanation of the data model used in WoSIS, I believe the section

could benefit from additional context. It's unclear whether users of the 2023 snapshot need a comprehensive understanding of the data model to utilize the data effectively. Therefore, I suggest explaining to data users how the data model impacts them. It would be helpful to include a figure or table, as the topic is quite technical and may require some abstraction. It's important to consider that your data users may not be database experts.

Good point, thanks. While the actual data model itself is not of direct concern to data users, it is critical for WoSIS staff to be able to efficiently and securely handle and process the shared data, also for versioning control. Importantly, based on the new data model the shared, standardised data can be served more easily, and in a wider range of formats, by ISRIC to the user community. To facilitate users, however, we preserved the file format of *wosis_latest* respectively the snapshot.

We have explicitly decided not to show/describe the full ISO data model (or add a Figure for this in the manuscript) as this would be over complicated (i.e. illegible), and not directly relevant to end-users of the 2023-snapshot. Alternatively, we have added some text for clarification, with reference to the technical documentation (see below). Key for users is that they have easy access to the derived data and related documentation. Sub 2.2. we added:

Main features of interest are the dataset (describes source of data), site (geo-spatial location where a soil investigation took place) and profile (sequence of pedo-genetic horizons along the depth of the profile or fixed-depth layers). The key modification in the new data model vis à vis the previous data model (Ribeiro *et al.* 2020) is the conditioning of analytical methods to the observation (see https://git.wur.nl/isric/databases/wosis-docs, last access: 24 April 2024). Changes made to the database schema and data over time were tracked using a migration tool (https://github.com/graphile/migrate, last access 24 April 2024). It maintains a record of the history, state, and dependencies of the database, including the conversion to the new data model.

You mention that the cleaned and standardized (processed) data are copied into the WoSIS database and subsequently removed from the staging area. However, it's unclear what happens to the source raw data. Is it still accessible, or is only the processed data published? I would appreciate more information on this aspect, including whether there is any form of data versioning in place. Additionally, while discussing data versioning, please elaborate on how you document the data processing steps. For example, is there a script stored along with the data, and is it released alongside the data as well?

The raw source data are always preserved "as is" in the ISRIC WDC-Soils repository, together with the data sharing agreements. For clarification, we added the following to p. 7, sub 2.3:

(note: the actual source data are permanently preserved in the data repository).

No scripts are attached with the data we distribute, The processing chain, including versioning, is managed within our institutional database itself. Concerning versioning, we added the following text sub. 2.2 (as indicated above):

Changes made to the database schema and data over time were tracked using a migration tool (https://github.com/graphile/migrate, last access 24 April 2024). It maintains a record of the history, state, and dependencies of the database, including the conversion to the new data model.

You mention that the data model was improved and new ETL procedures were developed, resulting in changes to the workflow. However, it's not clear why these changes were necessary. I suggest providing at least a brief explanation of why these changes were deemed necessary and how they contribute to enhancing WoSIS. It's important to illustrate how these changes benefit data users. For instance, what advantages do these developments offer? Additionally, how do the new ETL procedures differ from the previous ones? Given that various data integrity checks were already in place, what aspects are genuinely new, and why are they significant for users of the 2023 snapshot? Furthermore, how is the ETL procedure checklist managed? Is there a published document outlining this process? Providing more information and context for your readers would be beneficial.

Thanks, part of this query has been answered in response to your earlier queries above, see Section 2.1. The changes were mainly needed for the WoSIS database manager to efficiently handle a database of ever increasing complexity, in line with evolving international standards. Further, with these improvements the process of updating *wosis_latest* and generating future snapshots is much more efficient. That being said, the data screening procedures themselves are essentially the same. The (evolving) documentation, as already cited in the manuscript in Section 2.2 (de Sousa *et al.* , 2023), is available at: https://git.wur.nl/isric/databases/wosis-docs.

There is no published document yet that outlines the refactored ETL steps. However, we maintain an internal GIT page that describes the successive data screening stages to guide our data processors during the "ET phase" of the ETL process. We intend to make these procedures public once we have developed, and tested, the front-end for the ETL, which will allow data providers themselves to submit their own data set for consideration in WoSIS (see Section 8 Conclusions, paragraph 2).

Overall, I find this section to be overly technical, which may not be informative for data users unfamiliar with databases and data processing. Additionally, it could be enhanced by clearly highlighting the effective changes or improvements compared to the previous version. For instance, did the list of standardized soil properties remain the same, or were there any modifications? If so, why were these changes made? Are there any new data provisioning forms, or are the same channels as before still being utilized?

Furthermore, I recommend providing URLs for the assets in the text to facilitate access to the data. Ensuring easy access to information is crucial for users.

Assets mentioned in the text can all be accessed through the URLs provided with the related references. This was originally done because some URLs are rather long. Nonetheless, as recommended, for easier access the URLs have now been added in-text.

The list of soil properties (codes) remains the same, but the descriptions have been slightly changed to accommodate the ISO domain model conventions (i.e., property and procedure, See Appendix A).

You mention that all datasets shared with ISRIC are initially registered in the ISRIC Data Repository along with their metadata. However, it's unclear whether the metadata is open. This question arises from the explanation that third parties seeking access to "restricted" datasets would need to contact the source data provider. Accessing metadata would enable identification of restricted datasets and their owners. I recommend providing more information on this aspect, including the URL of the ISRIC Data Repository where one can find the source raw data (and possibly the metadata) with ISRIC.

As indicated, all datasets shared with our centre are registered in the ISRIC WDC-Soils with their metadata and licence, in order to enforce the security rules within WoSIS. The repository itself is only accessible to a limited number of ISRIC staff, using tokens. Our licences with data providers do not include the provision that ISRIC may redistribute any datasets that have been shared with us. Hence, the original text about 'seeking access to "restricted" datasets' was ill-phrased and it has been removed accordingly. Generalised information about our data providers, and possible data access to "open" sources, is provided at: https://www.isric.org/explore/wosis/wosis-contributing-institutions-and-experts.

**Data screening, quality control and standardisation**

Please specify if there are any changes in the consistency checks compared to the previous version.

The same checks are used in the refactored ETL workflow (note: we changed "new" ETL to "refactored" ETL). Alternatively, it should be noted that the plausibility checks (min, max) can change as more data become available for so far under-represented properties and our insight grows. This is common procedure in large soil databases (see e.g. https://www.icpforests.org/pdf/manual/2020/ICP_Manual_part16_2020_QAQC_Labs_version_2020-1.pdf). We added the following text to Section 3.3., p. 10, around line 20:

the plausibility limits themselves may evolve as more data become available for so far under-represented soil observations (see e.g., ICP Forests 2020, p. 25) and appropriate PostgreSQL "trigger mechanisms" have been implemented for this.

Please specify if there are any changes in the screening for duplicate profiles compared to the previous version.

Screening for duplicate profiles was especially important during the early stages of WoSIS when many datasets ingested were compilations of different data sources. These procedures have not changed, only their implementation. However, the year of sampling is now considered explicitly during the screening so as to permit handling of data from successive soil monitoring rounds (e.g. LUCAS topsoil surveys, and ICPF Forest data).

Please describe how the list shown at the top of page 10 differs from the content of subsection "2.4 Soil properties standardised".

Thanks, an oversight, it is actually the same list hence repetitive. The list has been maintained in Section 3.3 where it fits best. Alternatively, part of the original introduction (p. 2, line 25 to p.3 line7) was moved to Section 2.4 renamed as "Standardising soil property names", where it now fits better in view of other editorial comments made by RC1 concerning the "historical perspective".

You provide three measures for fitness-for-intended-use. How is this different from the previous snapshot?

These are the same measures as for the preceding (2019) snapshot. However, importantly, a change has been that we no longer use the term "geographical accuracy", which was found to be misleading after discussion with our Geo-experts. Instead, the term "positional uncertainty" is used now, as described in the manuscript sub 3.4.1.

**Spatial distribution of soil profiles and number of observations**

I think that you should present a map with the spatial distribution of samples of previous snapshot as well. This would give a better idea of the gains with the current snapshot.

This is a valuable recommendation as it complements the numbers in Table 1. We have added "profile density plots" to the paper to visualize changes between snapshots over

time in Section 4, which has now been re-worked into Sect. 4,1 "Spatial distribution and Sect 4.2 "Number of observations".

[Figure]

Further, some pieces of the original text (e.g., last paragraph Sect. 4) have been moved to the discussion section (Sect. 6) as this proved more appropriate. Such changes will

become apparent once the "track change" version of the revised manuscript has been submitted, following incorporation of the comments from reviewer 2.

**References cited in response to RC1:**

Ballabio C, Panagos P and Monatanarella L 2016. Mapping topsoil physical properties at European scale using the LUCAS database. *Geoderma* 261, 110-123. https://doi.org/10.1016/j.geoderma.2015.07.006

Batjes NH 2016. Harmonised soil property values for broad-scale modelling (WISE30sec) with estimates of global soil carbon stocks. *Geoderma* 269, 61-68. http://dx.doi.org/10.1016/j.geoderma.2016.01.034

Batjes NH and Bridges EM 1994. Potential emissions of radiatively active gases from soil to atmosphere with special reference to methane: development of a global database (WISE). *Journal of Geophysical Research* 99(D8), 16479-16489. http://dx.doi.org/10.1029/93JD03278

Batjes NH, Bridges EM and Nachtergaele FO 1995. World Inventory of Soil Emission Potentials: development of a global soil data base of process-controlling factors. In: Peng S, KT Ingram, HU Neue and LH Ziska (editors), *Climate Change and Rice*. Springer-Verlag, Heidelberg, pp 102-115.

Batlle-Bayer L, Batjes NH and Bindraban PS 2010. Changes in organic carbon stocks upon land use conversion in the Brazilian Cerrado: A review. *Agriculture, Ecosystems & Environment* 137, 47-58. http://dx.doi.org/10.1016/j.agee.2010.02.003

de Sousa LM 2023. *New WoSIS data model*, ISRIC- World Soil Information (Git page), Wageningen. https://git.wur.nl/isric/databases/wosis-docs

de Sousa LM, Calisto L, van Genuchten P, Turdukulov U and Kempen B 2023. *Data model for the ISO 28258 domain model*, ISRIC- World Soil Informatiom. https://iso28258.isric.org/

Dijkshoorn JA, Huting JRM and Tempel P 2005. *Update of the 1:5 million Soil and Terrain Database for Latin America and the Caribbean (SOTERLAC, ver. 2.0)*. Report 2005/01, ISRIC- World Soil Information, Wageningen. https://www.isric.org/documents/document-type/isric-report-200501-update-15-million-soil-and-terrain-database-latin

FAO 1989. *FAO-ISRIC Soil Database (SDB)*. World Soil Resources Report 60 (Reprinted), Food and Agriculture Organization of the United Nations, Rome

FAO 1990. *Guidelines for soil description (3rd rev. ed.)*, FAO, Rome, 45 p

FAO 2006. *Guidelines for soil description (Fourth ed.)*, FAO, Rome, 97 p. http://www.fao.org/docrep/019/a0541e/a0541e.pdf

FAO, ISRIC and UG 2007. *Soil and terrain database for central Africa (Burundi and Rwanda 1:1 million scale; Democratic Republic of the Congo 1:2 million scale)*. Land and Water Digital Media Series 33, Food and Agricultural Organization of the United Nations, ISRIC- World Soil Information and Universiteit Gent, Rome. https://www.isric.org/sites/default/files/isric_report_2006_07.pdf

FAO, ISRIC, UNEP and CIP 1998. *Soil and terrain digital database for Latin America and the Caribbean at 1:5 million scale*. Land and Water Digital Media Series No. 5, Food and Agriculture Organization of the United Nations, Rome

FAO, IIASA, ISRIC, ISSCAS and JRC 2012. *Harmonized World Soil Database (version 1.2)*, Prepared by Nachtergaele FO, van Velthuizen H, Verelst L, Wiberg D, Batjes NH, Dijkshoorn JA, van Engelen VWP, Fischer G, Jones A, Montanarella L., Petri M, Prieler S, Teixeira E and Xuezheng Shi. Food and Agriculture Organization of the United Nations (FAO), International Institute for Applied Systems Analysis (IIASA), ISRIC- World Soil Information, Institute of Soil Science- Chinese Academy of Sciences (ISSCAS), Joint Research Centre of the European Commission (JRC),

Laxenburg, Austria. http://webarchive.iiasa.ac.at/Research/LUC/External-World-soil-database/HWSD_Documentation.pdf

Hengl T, de Jesus JM, Heuvelink GBM, Gonzalez MR, Kilibarda M, Blagotic A, Shangguan W, Wright MN, Geng XY, Bauer-Marschallinger B, Guevara MA, Vargas R, MacMillan RA, Batjes NH, Leenaars JGB, Ribeiro E, Wheeler I, Mantel S and Kempen B 2017. SoilGrids250m: Global gridded soil information based on machine learning. *PLoS One* 12 https://doi.org/10.1371/journal.pone.0169748

ICP Forests 2020. *ICP Forests monitoring Manual. Part XVI: Quality assurance and control in laboratories (ver 2020-1)*, Eberswalde, Germany, 46 p. https://www.icp-forests.org/pdf/manual/2020/ICP_Manual_part16_2020_QAQC_Labs_version_2020-1.pdf

ICP Forests 2021. *ICP Forests monitoring Manual. Part X: Sampling and analysis of soil*, Eberswalde, Germany. https://storage.ning.com/topology/rest/1.0/file/get/9995584862?profile=original

Leenaars JGB, van Oostrum AJM and Ruiperez Gonzalez M 2014. *Africa Soil Profiles Database: A compilation of georeferenced and standardised legacy soil profile data for Sub Saharan Africa (version 1.2)*. Report 2014/01, Africa Soil Information Service (AfSIS) and ISRIC- World Soil Information, Wageningen, 160 p. http://www.isric.org/sites/default/files/isric_report_2014_01.pdf

Leenaars JGB, Claessens L, Heuvelink GBM, Hengl T, Ruiperez González M, van Bussel LGJ, Guilpart N, Yang H and Cassman KG 2018. Mapping rootable depth and root zone plant-available water holding capacity of the soil of sub-Saharan Africa. *Geoderma* 324, 18-36. https://doi.org/10.1016/j.geoderma.2018.02.046

Malhotra A, Todd-Brown K, Nave LE, Batjes NH, Holmquist JR, Hoyt AM, Iversen CM, Jackson RB, Lajtha K, Lawrence C, Vinduskova O, Wieder W, Williams M, Hugelius G and Harden J 2019. The landscape of soil carbon data: emerging questions, synergies and databases. *Progress in Physical Geography-Earth and Environment* 43, 707-719. https://doi.org/10.1177/0309133319873309

Oldeman LR and van Engelen VWP 1993. A World Soils and Terrain Digital Database (SOTER)- An improved assessment of land resources. *Geoderma* 60, 309-335.

Orgiazzi A, Ballabio C, Panagos P, Jones A and Fernandez-Ugalde O 2018. LUCAS Soil, the largest expandable soil dataset for Europe: a review. *European Journal of Soil Science* 69, 140-153. https://doi.org/10.1111/ejss.12499

Poggio L, de Sousa L, Batjes NH, Heuvelink GBM, Kempen B, Riberio E and Rossiter D 2021. SoilGrids 2.0: producing soil information for the globe with quantified spatial uncertainty. *SOIL* 7, 217–240. https://doi.org/10.5194/soil-7-217-2021

Ribeiro E, Batjes NH and Van Oostrum AJM 2020. *World Soil Information Service (WoSIS) - Towards the standardization and harmonization of world soil data. Procedures Manual 2020*. ISRIC Report 2020/01, ISRIC- World Soil Information, Wageningen, 153 p. http://dx.doi.org/10.17027/isric-wdc-2020-01

Ribeiro E, Batjes NH, Leenaars JGB, Van Oostrum AJM and Mendes de Jesus J 2015. *Towards the standardization and harmonization of world soil data: Procedures Manual ISRIC World Soil Information Service (WoSIS version 2.0)* Report 2015/03, ISRIC- World Soil Information, Wageningen, 110 p. http://www.isric.org/sites/default/files/isric_report_2015_03.pdf

Soil Survey Division Staff 1993. *Soil survey manual*, Handbook 18. Soil Conservation Service, U.S. Department of Agriculture, Washington, 503 p. https://www.nrcs.usda.gov/wps/portal/nrcs/detail/soils/ref/?cid=nrcs142p2_054262

Tempel P, van Kraalingen D, Mendes de Jesus J and Reuter HI 2013. *Towards an ISRIC World Soil Information Service (WOSIS ver. 1.0)*. ISRIC Report 2013/02, ISRIC- World Soil Information, Wageningen, 188 p. https://www.isric.org/sites/default/files/isric_report_2013_02.pdf

Turek ME, Poggio L, Batjes NH, Armindo RA, de Jong van Lier Q, de Sousa L and Heuvelink GBM 2022. Global mapping of volumetric water retention at 100, 330 and 15 000 cm suction using the WoSIS database. *International Soil and Water Conservation Research* https://www.sciencedirect.com/science/article/pii/S2095633922000636

van de Ven T and Tempel P 1994. *ISIS 4.0 - ISRIC Soil Information System: User Manual*. Technical Paper 15 (rev. ed.), International Soil Reference and Information Centre, Wageningen. https://www.isric.org/sites/default/files/ISRIC_TechPap15b.pdf

van Engelen VWP, Verdoodt A, Dijkshoorn K and van Ranst E 2006. *SOTER database for Central Africa -- DR Congo, Burundi and Rwanda (SOTERCAF; ver. 1.0)*. ISRIC REport 2006/07, Laboratory of Soil Science (University of Ghent), FAO and ISRIC- World Soil Information, Wageningen (http://www.isric.org/Isric/Webdocs/Docs/ISRIC_Report_2006_07.pdf; accessed 15 August 2007), 28 p

von Haden AC, Yang WH and DeLucia EH 2020. Soils' dirty little secret: Depth-based comparisons can be inadequate for quantifying changes in soil organic carbon and other mineral soil properties. *Global Change Biology* n/a https://doi.org/10.1111/gcb.15124

Wang M, Guo X, Zhang S, Xiao L, Mishra U, Yang Y, Zhu B, Wang G, Mao X, Qian T, Jiang T, Shi Z and Luo Z 2022. Global soil profiles indicate depth-dependent soil carbon losses under a warmer climate. *Nature Communications* 13, 5514. https://doi.org/10.1038/s41467-022-33278-w

---

## Author Comment (AC2)

**Response to Reviewer 2 (essd-2024-14-RC1):**

Note: Our responses to Revier2 are given below in blue.

I have reviewed the ESSD paper entitled "Providing quality-assessed and standardised soil data to support global mapping and modelling (WoSIS snapshot 2023)".

The paper is well written and provides a comprehensive description and technical guidance on the available data. As it is a continuation of previous "snapshots", the documentation base is solid and it is nice to see the continuous improvement by Dr Batjes and his team in terms of quality assurance and expansion of the dataset. My comments are minor, also because reviewer 1 has already provided excellent feedback and the authors have responded to these issues (which I will not repeat unless my opinion on the comment or feedback differs). The large number of profiles added is impressive, but I think the paper could benefit from being clearer about some things and highlighting potential problems.

- Uncertainty assessment. I like the idea of the multi-stage uncertainty assessment in terms of location, time of sampling, methods. However, given that the intended data users here will often not be soil scientists (which is good, soil data should be used), I think it is important to explain more clearly why these limitations are important. In addition, clear warnings should be given to non-soil scientists about the incomplete coverage of certain land uses, and about soil data from the global south (certain sub-Saharan African or Arctic regions have not seen much improvement in data coverage since the last snapshot), or also information about which soil layers are covered (most data are probably still rich in topsoil data, but not in subsoil). If pedotransfer functions are used, it is important to know to what extent the profiles are genetically sound, or whether they are highly disturbed or not representative of a particular soil region. I know this is a lot to ask of the authors and you may disagree with what the purpose of this paper is, but I feel there is a risk that some soil profiles here may be interpreted as representative when we know they are clearly not, for the reasons and examples I have given above.

Thanks for these important comments. Keeping in mind the scope of the present paper, for clarification, we have added that the data served from WoSIS are based on the 'best available' data, openly shared with our centre. Many of these source data were collected using purposive sampling, hence not based on a probabilistic sampling design. We have added the following sentence to Section 4.3:

'Importantly, prospective data users should also realise that the point/profile data shared for consideration in WoSIS are largely based on purposive sampling. During such 'traditional' surveys, soil surveyors identify sample locations based on their knowledge of the survey area, desired level of detail (scale) and objective of the survey, for example detailed or exploratory surveys (FAO, 2006; IUSS Working Group WRB, 2022; Soil Survey Staff, 2017). Hence, such 'legacy' data are not based on a probabilistic sampling scheme as recommended for digital soil mapping (Brus et al., 2011; Brus, 2022; Cramer et al., 2019; Heuvelink et al., 2007).'

- Related to this point, the maps provided by the authors in response to a similar comment from reviewer 1 are a good first step towards more information about where we have soil data and

where we don't (at least in this database). However, I think more clarity could be provided by providing some sort of meta-analysis on such clear limitations as which regions can be mapped from, or whether soil profiles are reasonably representative of a region. In my opinion, the authors need to make it clear whether it makes sense to colour an area on these snapshot maps at all when we know that we only have a handful of profiles and no systematic soil data from these regions (anything yellow on the maps provided is essentially no data). Again, these limitations may be clear to soil scientists, but are often overlooked when the data are used by other research communities who will have a strong interest in soil data from these regions (which should be encouraged if these limitations are understood).

A sentence, and reference, has been added concerning the need to carefully consider the 'area of applicability' of the data:

'For example, large areas of the globe are still poorly represented in WoSIS ( yellow areas in Fig. 3). As indicated, this issue can only be remedied when a larger selection of datasets is shared for consideration in WoSIS.'

- Similar to the tables for the total number and spatio-temporal variation of profiles in the database, I would find it useful to have more information available in terms of surface vs. subsurface data, land use, climate zone or soil type.

We will add a table showing the 'maximum' of soil depth sampled per continent for the 2023-snapshot, using three depth classes (0-30, 30-60 and > 60 cm Table 5) for illustration.

- Data inaccessibility: I find it quite shocking how much data is still not freely available, even from well-funded regions such as the EU, where essentially all soil data production is funded by taxpayers, no matter what the opinion of the individual data producer may be. This is not the fault of the authors, of course, but perhaps WoSIS needs to think about a mechanism to enforce true open access to all data (I have no idea what this would be, but if tens of thousands of profiles are not fully accessible, something is wrong with the system and against the spirit of open access).

We fully agree with your point of view and have been struggling with this issue for years. However, realistically, ISRIC itself is too small to resolve this important issue alone. As now indicated in the conclusions, in principle the Global Soil Partnership, through its affiliation with UN-FAO, would be a possible forum to tackle this challenge.

- This is more a question of interest or something to consider for the future: How should the dataset be viewed given the growing discrepancy between the time of sampling and assessment of soil parameters and the reliability of the values for a modern user? As we know that soil properties change over time, does this mean that we need to 'phase out'

certain parameters from profiles where we know that they may be significantly different today than they were decades ago? On a related note, Table 3 shows that more than a quarter of all profiles have no date. I think that's almost as bad as not knowing where these profiles are. Are these data points worth keeping at all, or will they cause confusion over time?

Certainly, we have been considering this. Hopefully, we can still trace the age of some of the older profiles. Older profiles remain relevant for the more stable soil properties such as soil texture. Alternatively, for soil carbon content and pH changes can be rapid. Data from different time periods remain useful, for example when using machine learning in space and time for modelling soil organic carbon change (Heuvelink et al., 2021).

In principle, as World Data Centre, we do not discard any legacy data. However, as clearly stressed in the manuscript, data users should 'filter' the available data according to their fitness for the intended use(s) they envision. That is, carefully assess the 'area of applicability of any' prediction.

References cited:

Brus, D. J., Kempen, B., and Heuvelink, G. B. M.: Sampling for validation of digital soil maps, European Journal of Soil Science, 62, 394-407, https://bsssjournals.onlinelibrary.wiley.com/doi/abs/10.1111/j.1365-2389.2011.01364.x, 2011.
Brus, J.: Spatial sampling with R, Chapman and Hall R/C, New York, 2022.
Cramer, M. D., Wootton, L. M., van Mazijk, R., and Verboom, G. A.: New regionally modelled soil layers improve prediction of vegetation type relative to that based on global soil models, Diversity and Distributions, 25, 1736-1750, https://onlinelibrary.wiley.com/doi/abs/10.1111/ddi.12973, 2019.
FAO: Guidelines for soil description (4th ed.), FAO, Rome, 97 pp., http://www.fao.org/docrep/019/a0541e/a0541e.pdf, 2006.
Heuvelink, G. B. M., Brown, J. D., and van Loon, E. E.: A probabilistic framework for representing and simulating uncertain environmental variables, International Journal of Geographical Information Science, 21, 497-513, https://doi.org/10.1080/13658810601063951, 2007.
Heuvelink, G. B. M., Angelini, M. E., Poggio, L., Bai, Z. G., Batjes, N. H., van den Bosch, R., Bossio, D., Estella, S., Lehmann, J., Olmedo, G. F., and Sanderman, J.: Machine learning in space and time for modelling soil organic carbon change, European Journal of Soil Science, 72, 1607-1623, https://doi.org/10.1111/ejss.12998, 2021.
IUSS Working Group WRB: World Reference Base for soil resources 2022 - International soil classification system for naming soils and creating legends for soil maps, International Union of Soil Sciences, Vienna (Austria), 284 pp., https://www.isric.org/sites/default/files/WRB_fourth_edition_2022-12-18.pdf, 2022.
Soil Survey Staff: Soil Survey Manual (rev. ed.), edited by: Ditzler, C., Scheffe, K., and Monger, H. C., United States Agriculture Handbook 18, USDA, Washington, 2017.

---

## Editor Decision (ED1)

Dear authors of the WoSIS 2023 snapshot,

the reviewers were happy with the changes and explanations in the manuscript, thank you for taking up their comments thoroughly! However, they did not comment on the data product itself. Some testing (not all files, but various) revealed aspects that should be improved so that it is in fact easy to use by other researchers.

**General comment:** The snapshot data itself on the repository should stand on its own and be easy to use, even without the ESSD paper. This requires at least a "technical" description of the files, headers and units, as well as some rough methodology, whereas more information in greater detail is then part of the ESSD paper. Ideally, this information would be gathered in a readme file outside the zip where the data is. (Also, at the moment, in the readme you can find the information on where to get the zip. This does not make sense if the readme is inside.) Some of this information you provide in the appendix, but it should be put alongside the data (or in both places) and it is not yet comprehensive. The following comments should clarify some of the issues with the data description.

**Specific comments:**

1) Opening .tsv files fails because of differences in columns in the header compared to the main data body (tested by different people and different programming languages, using ' ', '\s', '\s+', '\t' as separator). Might this be due to blank spaces in the string entries? Please check and improve.

2) The data needs a clear description of
   a. the files and what they contain (e.g. a list and short description).
   b. the individual columns within each files
   c. the units of the columns (e.g. the upper_depth information is supposedly in cm?)
   d. the category values that can occur in the columns and what they mean (e.g. "f" in the organic_surface)

   At the moment, these aspects are partly present in Appendix B in the paper, but not complete, and not described sufficiently so that one can understand and use the data without searching further in the paper or on the website (and sometimes even then without success). Please put this information in the readme. An example how this could look like is a template by GFZ Data Services.

3) Additional: the wosis_202312.gpkg could benefit from classes or mean values instead of or additional to text for the positional uncertainty so that it can be filtered and sorted.

Please address these issues so that the snapshot can be used easily and reach its potential.

Best regards,

Sibylle Haßler

---

## Author Response (AR2)

**Manuscript essd-2024-14**

Dear Editor, Sibylle Haßler,

Thank you for your positive comments on our manuscript to which we have only added a reference to the revised Readme file as discussed below.
These changes have been highlighted in blue in the annotated version of the manuscript:
"An online Readme file, that includes links to two short tutorials, provides additional technical information (https://www.isric.org/sites/default/files/Readme_WoSIS_202312_v2.pdf)." (Section 5, line 12) respectively, at the top of Appendix B (p. 27): "Technical details are provided in a Readme file (https://www.isric.org/sites/default/files/Readme_WoSIS_202312_v2.pdf).".

Your main, and important, queries relate to the dataset itself. In particular its description and how the data can be easily imported. These comments relate to materials described on a GeoNetwork landing page. As such, we could not flag our changes using track changes. Therefore, we provide links to the modified texts as currently uploaded online:

a) The revised snapshot landing page can be viewed see :
https://data.isric.org/geonetwork/srv/eng/catalog.search#/metadraf/e50f84e1-aa5b-49cb-bd6b-cd581232a2ec

We revised/expanded the Readme file based on your comments, see:
https://www.isric.org/sites/default/files/Readme_WoSIS_202312_v2.pdf. It includes responses to your queries about "a. the files and what they contain (e.g. a list and short description); b. the individual columns within each file; c. the units of the columns (e.g. the upper_depth information is supposedly in cm?) ; and d. the category values that can occur in the columns and what they mean (e.g. "f" in the organic_surface)."

b) To facilitate users, we have prepared two tutorials showing how to import TSV data into R programmes and Excel spreadsheets, respectively.
- https://www.isric.org/sites/default/files/Read_wosis_snapshot_2023_tsv_files_in_R.pdf
- https://www.isric.org/sites/default/files/Import_wosis_snapshot_2023_TSV_file_into_Excel.pdf

The new tutorials have also been added to the FAQ page: https://www.isric.org/explore/wosis/faq-wosis#Are_there_any_tutorials .

c) The Readme file is now directly accessible under 'Downloads and links'. It is also included in the zip file as, in our experience, users like to have the Readme file together with the data in their 'download' folder.

Further, you indicated 'the wosis_202312.gpkg could benefit from classes or mean values instead of or additional to text for the positional uncertainty so that it can be filtered and sorted.' Positional uncertainty in the snapshot is provided as four classes (text) that can be easily filtered. Since positional_uncertainty inherently is a 'coarse measure' providing mean values for a given class would not be useful as we do not know the 'true accuracy' of the coordinates (see section 3.4.1 in Manuscript).

We hope these replies satisfactorily answer your questions.

Best regards,

Niels Batjes on behalf of all co-authors

---

## Author Response (AR3)

**Manuscript essd-2024-14**

Dear Editor, Sibylle Haßler,

Thank you for noticing the defective URL that should have referred to a short Excel tutorial. We have updated the corresponding URL. Further, we have added a DOI, minted by DataCite, to the overarching Readme file (see https://doi.org/10.17027/isric-wdc-rgkv-d111).

We will take your final suggestion, considering the 'value averages', into consideration when the next WoSIS snapshot paper will be prepared.

Best regards,

Niels Batjes on behalf of all co-authors